# Dynamic closed states of a ligand-gated ion channel captured by cryo-EM and simulations

Urška Rovšnik[1], Yuxuan Zhuang[1], Björn O Forsberg[1,2], Marta Carroni[1], Linnea Yvonnesdotter[1], Rebecca J Howard[1] ⓘ, Erik Lindahl[1,3] ⓘ

**Ligand-gated ion channels are critical mediators of electrochemical signal transduction across evolution. Biophysical and pharmacological characterization of these receptor proteins relies on high-quality structures in multiple, subtly distinct functional states. However, structural data in this family remain limited, particularly for resting and intermediate states on the activation pathway. Here, we report cryo-electron microscopy (cryo-EM) structures of the proton-activated *Gloeobacter violaceus* ligand-gated ion channel (GLIC) under three pH conditions. Decreased pH was associated with improved resolution and side chain rearrangements at the subunit/domain interface, particularly involving functionally important residues in the β1–β2 and M2–M3 loops. Molecular dynamics simulations substantiated flexibility in the closed-channel extracellular domains relative to the transmembrane ones and supported electrostatic remodeling around E35 and E243 in proton-induced gating. Exploration of secondary cryo-EM classes further indicated a low-pH population with an expanded pore. These results allow us to define distinct protonation and activation steps in pH-stimulated conformational cycling in GLIC, including interfacial rearrangements largely conserved in the pentameric channel family.**

## Introduction

Pentameric ligand-gated ion channels are major mediators of fast synaptic transmission in the mammalian nervous system and serve a variety of biological roles across evolution (1). Representative X-ray and cryo-electron microscopy (cryo-EM) structures in this family have confirmed a fivefold pseudosymmetric architecture, conserved from prokaryotes to humans (2). The extracellular domain (ECD) of each subunit contains β-strands β1–β10, with the characteristic Cys- or Pro-loop (3) connecting β6–β7, and loops A–F enclosing a canonical ligand-binding site (4) at the interface between principal and complementary subunits. The transmembrane domain (TMD) contains α-helices M1–M4, with M2 lining the channel pore, and an intracellular domain of varying length (2–80 residues) inserted between M3 and M4. Extracellular agonist binding is thought to favor subtle structural transitions from resting to intermediate or "flip" states (2), opening of a transmembrane pore (5), and in most cases, a refractory desensitized phase (6). Accordingly, a detailed understanding of pentameric channel biophysics and pharmacology depends on high-quality structural templates in multiple functional states. However, high-resolution structures can be biased by stabilizing measures such as ligands, mutations, and crystallization, leaving open questions as to the wild-type activation process.

As a model system in this family, the *Gloeobacter violaceus* proton-gated ion channel (GLIC) has historically offered both insights and limitations (7). This prokaryotic receptor has been functionally characterized in multiple cell types (8) and crystallizes readily under activating conditions (pH ≤ 5.5) (9, 10), producing apparent open structures up to 2.22 Å resolution (11) in the absence and presence of various ligands (12, 13, 14, 15, 16, 17, 18, 19, 20, 21) and mutations (22, 23, 24, 25). Additional low-pH X-ray structures of GLIC have been reported in lipid-modulated (26) and the so-called locally closed states (27, 28, 29, 30), with a hydrophobic constriction at the pore midpoint (I233, I9′ in prime notation) as predicted for closed channels throughout the family (31). Crystallography at neutral pH has also been reported, but only to relatively low resolution (4.35 Å), suggesting a resting state with a relatively expanded, twisted ECD as well as a contracted pore (32, 33). Alternative structural methods have supported the existence of multiple nonconducting conformations (34, 35, 36), and biochemical studies have implicated titratable residues including E35 and E243 in pH sensing (11, 25, 36, 37). However, due in part to limited structural data for wild-type GLIC in resting, intermediate, or desensitized states, the mechanism of proton gating remains unclear.

Here, we report single-particle cryo-EM structures and molecular dynamics (MD) simulations of GLIC at pH 7, 5, and 3. Taking advantage of the relatively flexible conditions accessible to cryo-EM, we resolve

[1]Department of Biochemistry and Biophysics, Science for Life Laboratory, Stockholm University, Solna, Sweden   [2]Division of Structural Biology, Wellcome Centre for Human Genetics, University of Oxford, Oxford, UK   [3]Department of Applied Physics, Science for Life Laboratory, Kungliga Tekniska Högskolan Royal Institute of Technology, Solna, Sweden

Correspondence: erik.lindahl@scilifelab.se

multiple closed structures, distinct from those previously reported by crystallography. We find rearrangements of E35 and E243 differentiate deprotonated versus protonated conditions, providing a dynamic rationale for proton-stimulated remodeling. Classification of cryo-EM data further indicated a minority population with a contracted ECD and expanded pore. These results support a dissection of protonation and activation steps in pH-stimulated conformational cycling, by which GLIC preserves a general gating pathway via interfacial electrostatics rather than ligand binding.

# Results

### Differential resolution of GLIC cryo-EM structures with varying pH

To characterize the resting state of the prokaryotic pentameric channel GLIC, we first obtained single-particle cryo-EM data under resting conditions (pH 7), resulting in a map to 4.1 Å overall resolution (Figs 1A and B, S1A–C, S2A, and S3A, and Table 1). Local resolution was between 3.5 and 4.0 Å in the TMD, including complete backbone traces for all four transmembrane helices. Side chains in the TMD core were clearly resolved (Fig S4A), including a constriction at the I233 hydrophobic gate (I9′, 2.9 Å Cβ-atom radius), consistent with a closed pore. Whereas some extracellular regions were similarly well resolved (Fig S4B), local resolution in the ECD was generally lower (Fig 1B), with some atoms that could not be definitively built in the β1–β2 loop, β8–β9 loop (loop F), and at the apical end of the ECD (Fig 2B).

GLIC has been thoroughly documented as a proton-gated ion channel, conducting currents in response to low extracellular pH with half-maximal activation around pH 5 (8). Taking advantage of the flexible buffer conditions accessible to cryo-EM, we obtained additional reconstructions under partial and maximal (pH 5 and pH 3) activating conditions, producing maps to 3.4 and 3.6 Å, respectively (Figs 1C and D, S1A–C, S2B and C, and S3B and C). Overall map quality improved at lower pH, although local resolution in the TMD remained high relative to the ECD (Fig 1C and D). As a partial check for our map comparisons, we also selected random subsets containing equivalent numbers of particles from each dataset; we found the pH-5 and pH-3 datasets still produced higher quality reconstructions than those at pH 7 (Fig S5A–C). This comparison indicates that differential resolution could not be trivially attributed to data quantity, although it does not identify the cause of differential data quality. Surprisingly, backbone alignments of models at both pH 5 and pH 3 indicated close fits to the pH-7 model (root mean-squared deviation over non-loop Cα atoms, Root mean-squared deviation [RMSD] ≤ 0.6 Å) in both the ECD and TMD, including a closed conformation of the transmembrane pore (Figs 1B–D and 2A). Interestingly, superposition of our cryo-EM reconstructions with equivalent molecules in an open-state GLIC crystal lattice (16) revealed clashes particularly at the outer periphery of the ECD (Fig S6), indicating packing effects that could influence conformational selection. Indeed, all three models deviated moderately from resting (protein data bank [PDB] ID: 4NPQ, ECD RMSD ≤ 1.4 Å, TMD RMSD ≤ 0.8 Å) but further from open X-ray structures (PDB ID: 4HFI, ECD RMSD ≤ 2.2 Å, TMD RMSD ≤ 1.9 Å),

suggesting systematic differences in EM versus crystallized conditions, as well as general alignment to a conserved closed-state backbone. Still, variations in local resolution and side chain orientation indicated pH-dependent conformational changes at the subunit–domain interface, as described below.

### Side chain rearrangements in low-pH structures

In the ECD, differential resolution was notable in the β1–β2 loop, particularly in the principal proton-sensor (11, 25) residue E35. At pH 7 and pH 5, little definitive density was associated with this side chain (Fig 2B, left, center); conversely at pH 3, it clearly extended towards the complementary loop F, forming a possible hydrogen bond with T158 (3.5 Å donor–acceptor; Fig 2B, right). Notably, this interaction mirrored that observed in open X-ray structures (Fig S7B), despite the general absence of open-like backbone rearrangements in the cryo-EM structure. At the midpoint of the same β1–β2 loop, density surrounding basic residue K33 was similarly absent at pH 7 and pH 5, but clearly defined a side chain oriented down towards the TMD at pH 3 (Fig 2B). An additional β1–β2 residue, D31, could also be uniquely built at pH 3, oriented in towards the central vestibule. Enhanced definition of these side chains could indicate a relatively uniform state of this loop at pH 3, improved cryo-EM signal for protonated versus anionic acidic groups, or other factors; dynamics in this region were accordingly explored in the next section. Among seven other acidic residues (E75, D97, D115, D122, D145, D161, and D178) associated with improved densities at low pH, only D122 has been shown to substantially influence channel properties (29); this residue is involved in an electrostatic network conserved across evolution, with substitutions decreasing channel expression as well as function (25), suggesting its role may involve assembly or architecture more than proton sensitivity.

In the TMD, rearrangements were observed particularly in the M2–M3 loop, a region thought to couple ECD activation to TMD-pore opening. At pH 7, K248 at the loop midpoint oriented down toward the M2 helix, where it could form an intrasubunit hydrogen bond with E243. Conversely, at pH 5 and pH 3, K248 reoriented out towards the complementary subunit. Residue K248 has been implicated in GLIC ECD-TMD coupling (27), while E243 was shown to be an important proton sensor (11); indeed, rearrangement of K248 to an interfacial orientation is also evident in open X-ray structures, with an accompanying iris-like motion of the M2–M3 region—including both K248 and E243—outward from the channel pore (Fig S7B). Thus, side chain arrangements in both the ECD and TMD were consistent with proton activation, while maintaining a closed pore.

### Remodeled electrostatic contacts revealed by molecular dynamics

To elucidate the basis for variations in local resolution (Fig 1B–D) and side chain orientation (Fig 2B–D) described above, and assess whether it is a property of the state or experiment, we ran quadruplicate 1 μs all-atom MD simulations of each cryo-EM structure, embedded in a lipid bilayer and 150 mM NaCl. To further test the role of pH, we ran parallel simulations with a subset of acidic residues modified to approximate the probable protonation pattern under activating conditions, as previously described (13). For comparison,

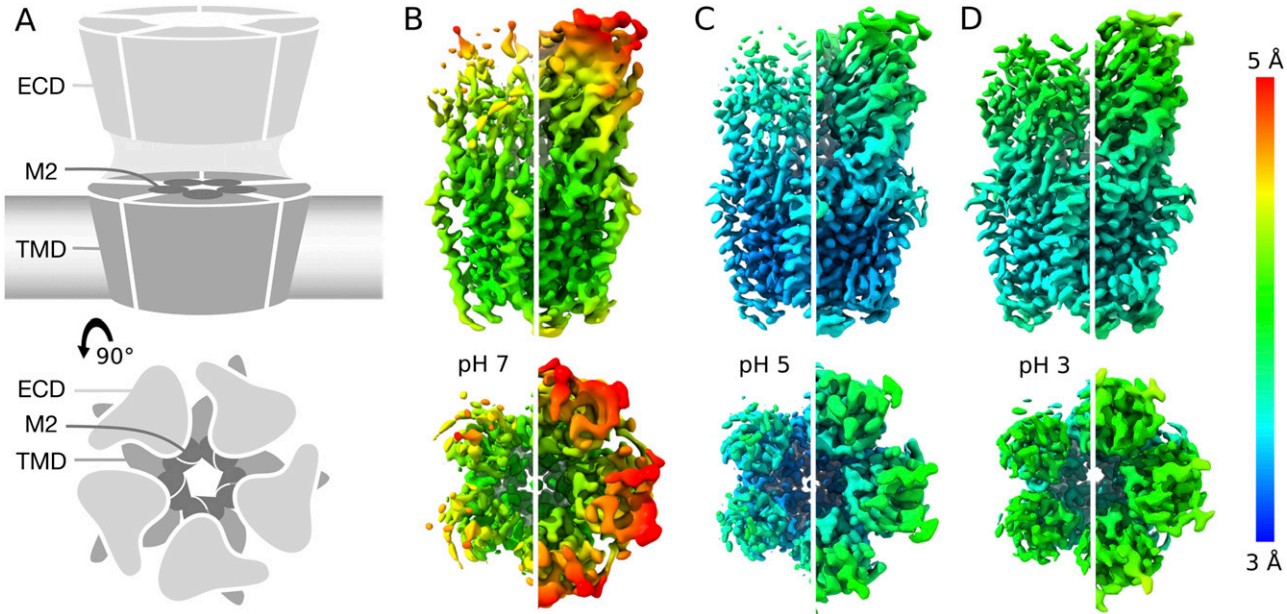

**Figure 1. Differential resolution of GLIC cryo-EM structures with varying pH.**
**(A)** Cartoon representations of GLIC, viewed from the membrane plane (top) or from the extracellular side (bottom). Pentameric rings represent the connected extracellular (ECD, light gray) and transmembrane (TMD, medium gray) domains, with the latter embedded in a lipid bilayer (gradient) and surrounding a membrane-spanning pore formed by the second helix from each subunit (M2, dark gray). **(A, B)** Cryo-EM density for the majority class (state 1) at pH 7 to 4.1 Å overall resolution, viewed as in panel (A) from the membrane plane (top) or from the extracellular side (bottom). Density is colored by local resolution according to scale bar at far right and contoured at both high (left) and low threshold (right) to reveal fine and coarse detail, respectively. **(B, C)** Density viewed as in panel (B) for state 1 at pH 5, reconstructed to 3.4 Å overall resolution. **(B, D)** Density as in panel (B) for state 1 at pH 3, reconstructed to 3.6 Å overall resolution.

X-ray structures reported previously under resting and activating conditions were also simulated, at neutral and low-pH protonation states, respectively. Simulation RMSD converged to a similar degree within 250 ns (Fig S8A), with all except the open X-ray structure dehydrated around the hydrophobic gate (Fig S8B). Simulations of all three cryo-EM structures exhibited elevated RMSD for the extracellular domains (RMSD < 3.5 Å) versus transmembrane regions (RMSD < 2.0 Å), consistent with higher flexibility in the ECD; both domains exhibited similarly low RMSD in simulations of the open X-ray structure (Fig S8A). Beyond these domain-level deviations, distinct transitions were observed with decreasing pH in a subset of side chain and backbone atoms at the ECD subunit interface and in the M2–M3 loop, as described below (Figs 3 and S8C).

In the ECD, simulations suggested a dynamic basis for pH-dependent interactions of the E35 proton sensor at the intersubunit β1–β2/loop-F interface (Fig 3A–C). Under resting (deprotonated) conditions, negatively charged E35 attracted cations from the extracellular medium, forming a direct electrostatic contact with Na$^+$ in >35% of simulation frames (Fig 3A and B). These environmental ions were not coordinated by other protein motifs in a rigid binding site, potentially explaining poorly resolved densities in this region in neutral-pH structures. Cation coordination decreased slightly in the pH-3 structure even under deprotonated conditions, but was effectively eliminated in all simulations under activating (protonated) conditions. In parallel, mean Cα-distances between E35 and the complementary T158 contracted in protonated simulations to values approaching the open X-ray structure (Fig 3A and C), as the now-uncharged glutamate released Na$^+$ and became

available to interact with the proximal threonine. Interestingly, loop F—including T158—exhibited lower Cα deviations in simulations of the pH-3 versus higher pH models, a pattern also exhibited by low-versus neutral-pH X-ray structures (Fig S8C).

In the TMD, simulations further substantiated gating-like rearrangements in the M2–M3 loop (Fig 3D–F). In simulations of the pH-7 structure under deprotonated conditions, the K248 side chain was attracted down in each subunit towards the negatively charged E243; similar to the starting structure (Fig 2C and D), these residues formed an electrostatic contact in >70% of trajectory frames (Fig 3D and E). In simulations of the pH-3 structure, K248 more often oriented out toward the subunit interface (Fig 3D and E), also as seen in the corresponding structure (Fig 2C and D). Moreover, E243-K248 interactions decreased in protonated versus deprotonated simulations of all three structures, with the prevalence of this contact in protonated simulations at pH 3 (<25%) approaching that in open X-ray structures (Fig 3E). To explore larger scale rearrangements in M2–M3 that might accompany these side chain transitions, we further plotted the two lowest principal components (PC) of Cα atom Cartesian coordinate covariance in the M2–M3 loop, revealing distinct conformation distributions in simulations of pH-7, pH-5, and pH-3 models (Fig 3F). The two dominant PCs for this motif were associated with flipping of the central loop (Cα atoms near K248) from a downward to outward orientation (PC1), and stretching of the loop backbone across the subunit/domain interface (PC2). Projected along these axes, structures determined in decreasing pH conditions increasingly approximated the open X-ray structure, particularly in protonated simulations. Thus, in addition to

**Table 1.  Cryo-EM data collection and model refinement statistics.**

| Data collection and processing | pH-7 dataset | pH-5 dataset | pH-3 dataset |
|---|---|---|---|
| Microscope | FEI Titan Krios | FEI Titan Krios | FEI Titan Krios |
| Magnification | 165,000 | 165,000 | 165,000 |
| Voltage (kV) | 300 | 300 | 300 |
| Electron exposure ($e^-/Å^2$) | ~50 | ~50 | ~50 |
| Defocus range ($\mu m$) | 2.0–3.8 | 2.0–3.8 | 2.0–3.8 |
| Pixel size (Å) | 0.82 | 0.82 | 0.83 |
| Symmetry imposed | C5 | C5 | C5 |
| Number of images | ~5,300 | ~7,000 | ~6,400 |
| Particles picked | ~700,000 | ~1 million | ~690,000 |
| Particles refined | 86,201 | 351,643 | 214,463 |
| Refinement | | | |
| Initial model used | 4NPQ monomer | 4NPQ monomer | 4NPQ monomer |
| Resolution (Å) | 4.1 | 3.4 | 3.6 |
| FSC threshold | 0.143 | 0.143 | 0.143 |
| Map sharpening B-factor | −278 | −223 | −225 |
| Model composition | | | |
| Non-hydrogen protein atoms | 10,175 | 11,555 | 11,630 |
| Protein residues | 1,440 | 1,540 | 1,535 |
| Ligands | 0 | 0 | 0 |
| B-factor ($Å^2$) | 57 | 20 | 34 |
| Root mean-squared deviation | | | |
| Bond Lengths (Å) | 0.006 | 0.005 | 0.006 |
| Bond angles (°) | 0.616 | 0.599 | 0.664 |
| Validation | | | |
| MolProbity score | 1.87 | 1.93 | 1.77 |
| Clashscore | 10.73 | 9.66 | 6.12 |
| Poor rotamers (%) | 0 | 0 | 0 |
| Ramachandran plot | | | |
| Favored (%) | 95.4 | 93.7 | 93.4 |
| Allowed (%) | 4.6 | 6.3 | 6.6 |
| Outliers (%) | 0 | 0 | 0 |

substantiating differential stability in extracellular and transmembrane regions, MD simulations offered a rationale for dynamic pH-dependent rearrangements at the subunit/domain interface.

### Minority classes suggest alternative states

Compared to the best-quality reconstructions obtained at each pH (state 1, Fig 1B–D), cryo-EM data classification in all cases identified minority populations, indicating the presence of multiple conformations that could correspond to functionally relevant states. In particular, a minority class (state 2) representing ~23,000 (3.3%) of the particles used for 3D classification at pH 3 was visibly contracted and rotated in the ECD, relative to the majority class (state 1) (Figs S9A and S10A–C). Although a model could not be confidently

built at this resolution (4.9 Å), partial refinement of the state-1 backbone into the state-2 density at pH 3 revealed systematic reductions in ECD spread and domain twist, echoing transitions from resting to open X-ray structures (Fig S9B) (32, 33). Minority classes could also be reconstructed at pH 7 and pH 5, although to lower resolution (5.8 and 5.1 Å, respectively), and with less apparent divergence from state 1 in each condition (Fig S11A–C).

In the TMD, pH-3 state 2 also exhibited a tilted conformation of the upper M2 helices, outward towards the complementary subunit and away from the channel pore relative to state 1 (Figs 4A–C and S9A). Whereas the upper pore in state-1 models was almost indistinguishable from that of the resting X-ray structure (Figs 4 and S11A–C), in pH-3 state 2 it transitioned in the direction of the open X-ray state (Fig 4B). Static pore profiles (38) revealed expansion of

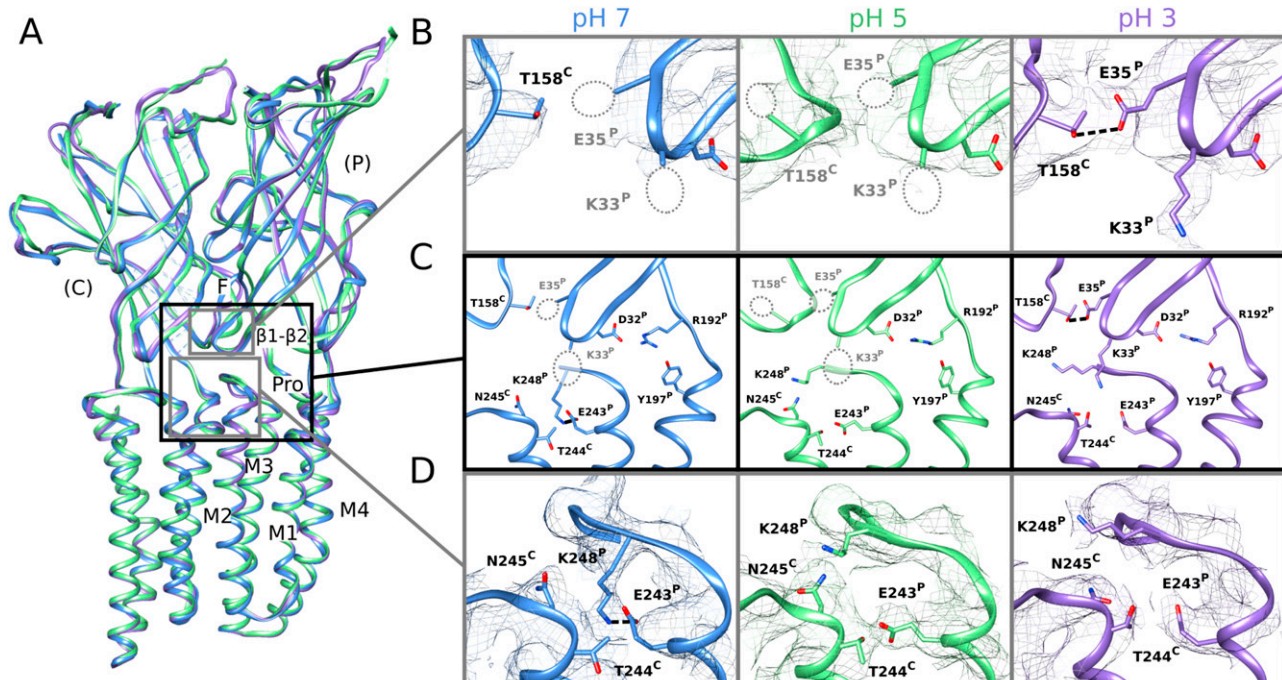

**Figure 2. Side chain rearrangements at subunit interfaces in low-pH structures.**
**(A)** Overlay of predominant (state 1) GLIC cryo-EM structures at pH 7 (blue), pH 5 (green), and pH 3 (lavender), aligned on the full pentamer. Two adjacent subunits are viewed as ribbons from the channel pore, showing key motifs including the β1–β2 and Pro loops and M1–M4 helices from the principal subunit (P), and loop F from the complementary subunit (C). **(A, B)** Zoom views of the upper gray-boxed region in panel (A), showing cryo-EM densities (mesh at σ = 0.25) and side chain atoms (sticks, colored by heteroatom) around the intersubunit ECD interface between a single principal β1–β2 loop and complementary loop F at each pH. As indicated by dotted circles, side chains including β1–β2 residues K33 and E35 could not be definitively built at pH 7 (left) or pH 5 (center) but were better resolved at pH 3 (right), including a possible hydrogen bond between E35 and T158 (dashed line, 3.2 Å). **(A, C)** Zoom views of the black-boxed region in panel (A), showing key side chains (sticks, colored by heteroatom) at the domain interface between one principal β1–β2, pre-M1, and M2–M3 region, and the complementary loop-F and M2 region. Dotted circles indicate side chains that could not be definitively built in the corresponding conditions; dashed lines indicate possible hydrogen bonds implicated here in proton-stimulated conformational cycling. Residues contributing to a conserved electrostatic network at the domain interface (D32, R192, Y197) are also shown. **(A, D)** Zoom views of the lower gray-boxed region in panel (A), showing cryo-EM densities (mesh) and side chain atoms (sticks, colored by heteroatom) around the intersubunit TMD interface between principal and complementary M2–M3 regions at each pH. A potential hydrogen bond between E243 and K248 at pH 7 (left, dashed line, 3.1 Å) is disrupted at pH 5 (center) and pH 3 (right), allowing K248 to reorient towards the subunit interface.

pH-3 state 2 at channel-facing residues S230–I240 (S6′–I16′) (Fig 4D). Although the open X-ray structure was substantially more expanded, MD simulations of that state consistently converged to a partly contracted pore at and above S6′; indeed, some open-state replicates sampled profiles overlapping pH-3 state 2 (Fig 4D), while remaining hydrated at the I9′ hydrophobic gate (Fig S8B). In contrast, simulations of state-1 cryo-EM and resting X-ray structures did not substantially contract in the upper pore (Fig S11D–H). Thus, minority classes indicated the presence of alternative functional states consistent with transition towards an open state at low pH.

## Discussion

Structures of GLIC in this work represent the first reported by cryo-EM, to our knowledge, covering multiple pH conditions and revealing electrostatic interactions at key subunit interfaces which are further substantiated by microsecond-scale MD simulations. Our data support a multi-step model for proton activation, in which closed states are characterized by a relatively flexible expanded

ECD and a contracted upper pore (Fig 5A). Protonation of both ECD (E35) and TMD (E243) glutamates relieves charge interactions associated with the resting state, enabling side chain remodeling particularly in the β1–β2 and M2–M3 loops, without necessarily altering the backbone fold (Fig 5B). Further rearrangements of the backbone are proposed to retain protonated side chain arrangements by contracting the ECD and expanding the TMD pore, as indicated both by a minority class in our low-pH cryo-EM data (Fig 4), and by comparisons with apparent open X-ray structures (Fig 5C).

Direct involvement of extracellular loops β1–β2 and F in proton sensing proved consistent with several recent predictions. Mutations at β1–β2 residue E35 were among the most impactful of any acidic residues in previous scanning experiments (25). Moreover, past spectroscopic studies showed the pH of receptor activation recapitulates the individual pKa of this residue, implicating it as the key proton sensor (11). In contrast, mutations at K33 have not been shown to dramatically influence channel function; indeed, previous crosslinking with the M2–M3 loop showed this position can either preserve or inhibit proton activation (27), suggesting the improved definition we observed for this side chain at low pH was more a byproduct of local remodeling than a determinant of gating. At E35's

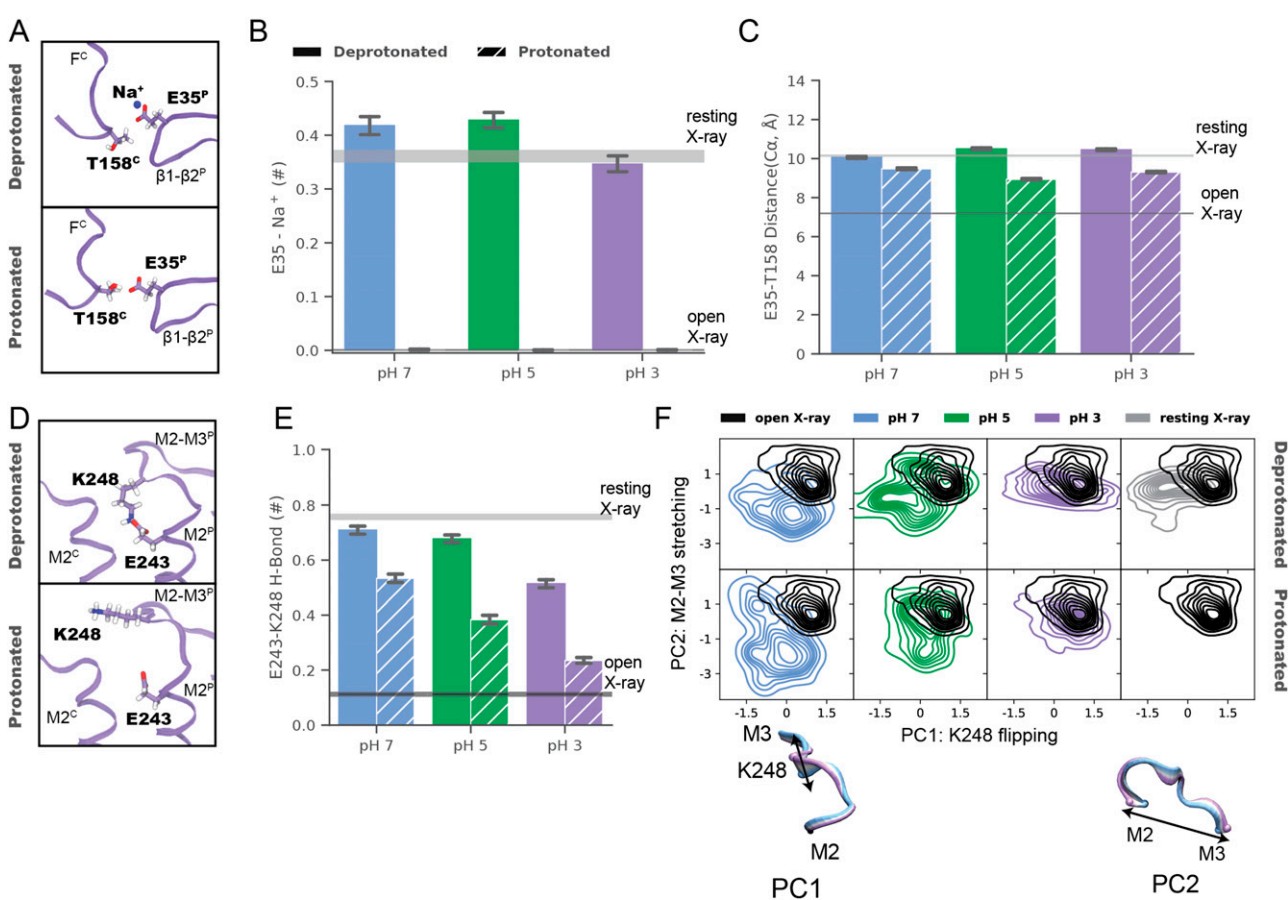

**Figure 3. Remodeled electrostatic contacts revealed by molecular dynamics.**
**(A)** Zoom views as in Fig 2B of the ECD interface between a single principal (P, right) β1–β2 loop and complementary (C, left) loop F (lavender ribbons) in representative snapshots from molecular dynamics simulations of the pH 3 (state 1) cryo-EM structure, with side chains modified to approximate resting (deprotonated, top) or activating (protonated, bottom) conditions. Depicted residues and proximal ions (sticks, colored by heteroatom) show deprotonated E35 in contact with $Na^+$, whereas protonated E35 interacts with T158. **(B)** Charge contacts between E35 and environmental $Na^+$ ions in simulations under deprotonated (solid) but not protonated (striped) conditions of state-1 cryo-EM structures determined at pH 7 (blue), pH 5 (green), or pH 3 (lavender). Histograms represent median ± 95% confidence interval over all simulations in the corresponding condition. Horizontal bars represent median ± confidence interval values for simulations of resting (gray) or open (black) X-ray structures. **(B, C)** Histograms as in panel (B) showing intersubunit Cα-distances between E35 and T158, which decrease in protonated (striped) versus deprotonated (solid) conditions. **(D)** Zoom views as in Fig 2D of the TMD interface between principal (P, right) and complementary (C, left) M2–M3 loops (lavender ribbons) in representative snapshots from simulations of the pH 3 (state 1) cryo-EM structure. Depicted residues (sticks, colored by heteroatom) show K248 oriented down towards E243 in deprotonated conditions (top), but out towards the subunit interface in protonated conditions (bottom). **(B, E)** Histograms as in panel (B) showing electrostatic contacts between E243 and K248, which decrease in pH-3 (lavender) versus pH-7 (blue) and pH-5 structures (green), and in protonated (striped) versus deprotonated (solid) simulation conditions. **(F)** Principal component (PC) analysis of M2–M3 loop motions in simulations under deprotonated (top) or protonated conditions (bottom) of state-1 cryo-EM structures determined at pH 7 (blue), pH 5 (green), and pH 3 (lavender). For comparison, simulations of previous resting (gray) and open (black) X-ray structures are shown at right, and open-structure results are superimposed in each panel. Inset cartoons illustrate structural transitions associated with dominant PCs (blue–lavender from negative to positive values), representing flipping of residue K248 (PC1) and stretching of the M2–M3 loop (PC2).

closest contact, loop-F residue T158, chemical labeling has been shown to reversibly inhibit activation (11), supporting a role in channel function. Interestingly, loop F adopted a different conformation in our structures at pH 5 compared to pH 7 or pH 3 (Fig 2B and C), suggesting this region samples a range of conformations; indeed, previous spin-labeling studies indicated this position, along with several neighbors on the β8 strand, to be highly dynamic (39). Although its broader role in pentameric channel gating remains controversial, loop F has often been characterized as an unstructured motif that undergoes substantial rearrangement during ligand binding (40), echoing the mechanism proposed here for GLIC.

Transmembrane residues E243 and K248 have been similarly implicated in channel function, albeit secondary to E35 in proton sensing. Residue E243 on the upper M2 helix is exposed to solvent, and has been predicted to protonate at low pH (13, 37). Previous studies have shown some mutations at this position to be silent, whereas others dramatically alter pH sensitivity (11, 25, 36, 41), suggesting its involvement in state-dependent interactions is complex. Interestingly, E243 has also been shown to mediate interactions with allosteric modulators via a cavity at the intersubunit interface (15), indicating a role for this residue in agonist sensitivity and/or coupling. At K248, cysteine substitution was previously shown to increase proton sensitivity (27), consistent with a weakening of charge interactions specific to the resting state (Fig 5). Past simulations based on X-ray structures also showed K248 to prefer intrasubunit interactions at rest, versus intersubunit interactions in

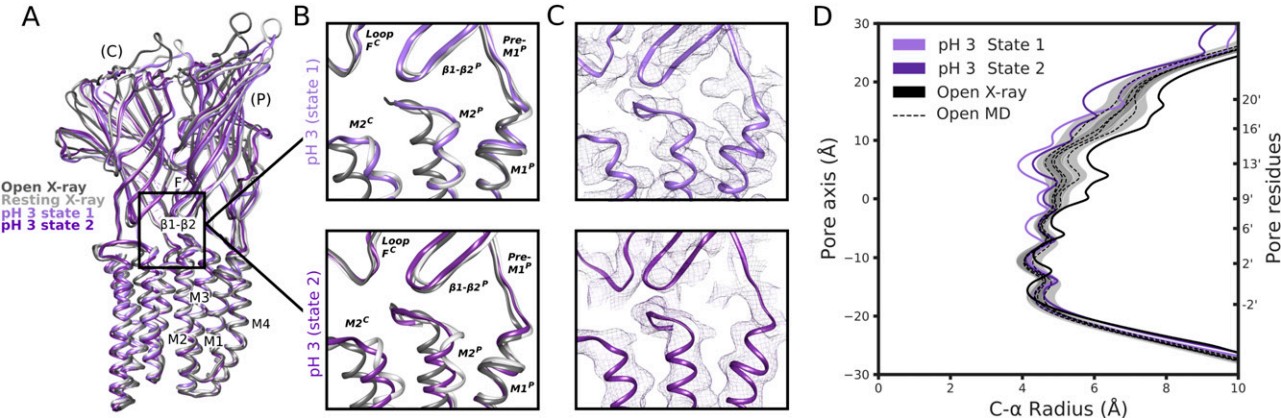

**Figure 4. Minority classes suggest alternative states.**
**(A)** Overlay as in Fig 2A of state-1 (lavender) and state-2 (purple) GLIC cryo-EM structures, along with apparent resting (white, PDB ID: 4NPQ) and open (gray, PDB ID: 4HFI) X-ray structures, aligned on the full pentamer. Adjacent principal (P) and complementary (C) subunits are viewed as ribbons from the channel pore. **(A, B)** Zoom views of the black-boxed region in panel (A), showing key motifs at the domain interface between one principal β1-β2, pre-M1, and M2–M3 region, and the complementary loop-F and M2 region, for resting (white) and open (gray) X-ray structures overlaid with pH-3 cryo-EM state 1 (top, lavender) or state 2 (bottom, purple). **(B, C)** Zoom views as in panel (B), showing cryo-EM densities (mesh) and backbone ribbons for pH-3 state 1 (top, lavender) or state 2 (bottom, purple). **(D)** Pore profiles (38) representing Cα radii for pH-3 cryo-EM state-1 (lavender) and state-2 (purple) structures, open X-ray (black) structure, and quadruplicate 1 μs molecular dynamics simulations of the open X-ray model (median, dashed black; 95% confidence interval, gray).

the open state (37), although E243/K248 interactions were particularly apparent in the present work.

Our reconstructions offer a structural rationale for the predominance of open and locally closed states in the crystallographic literature. The apparent resting state (pH 7) was characterized by relatively low reconstructed resolution (Figs 1B and 2A) and flexibility in the ECD (Figs S7A and 5A), particularly at the domain interface and peripheral surfaces, potentially conferring entropic favorability. Crystallization enforces conformational homogeneity, and may select for rigidified states particularly at crystal-contact surfaces (Fig S6); according to the model above (Fig 5), such conditions could bias towards a more uniform open state. Indeed, our simulations suggested the precise pore geometry of the open X-ray structure may not persist outside the crystal, potentially sampling more contracted conformations approaching pH-3 state 2 (Fig 4) while remaining generally hydrated (Fig S7B). Interestingly, a proposed desensitized-GLIC structure determined by co-crystallization with the inhibitor docosahexaenoic acid also contained a partly contracted upper pore (26), supporting the presence of alternative conformations in the conduction pathway. Conversely, cryo-EM could be expected to reveal favored but flexible conformations (Figs 2 and 4), with the caveat that there might instead be a bias towards higher resolution states. Thus, although the low resolution of our minority-class reconstructions precluded atomistic modeling or quantitative simulations of permeability, our pH-3 minority class could represent an intermediate or alternative state along the GLIC gating pathway. This state was associated with 3.3% of particles used for 3D classification at pH 3, with state 1 accounting for 46% (23,000 versus 214,000 out of 690,000, Figs S1C and S10C); however, we refrained from attributing particle counts directly to state distributions. Indeed, additional conformations including an open state are likely present in our samples, but were not clearly distinguished among predominant classes of well-ordered particles. To our knowledge, all cryo-EM classification methods are stochastic optimization tools subject to local minima, and not well suited to inferring quantitative ensemble statistics, particularly when the number and functional annotation of total states remains unclear.

Interestingly, the growing catalog of pentameric ligand-gated ion channel structures now includes several examples in which experimental data—largely from X-ray crystallography or cryo-EM—correspond to unanticipated functional states. In the case of *Erwinia Chrysanthemi* ligand-gated ion channel (ELIC), all structures to date have been incompatible with conduction even in the presence of saturating agonist, possibly because of a lack of critical lipid interactions (42, 43, 44). Some eukaryotic channels exhibit an opposite tendency, adopting hydrated or open states even where prolonged exposure to agonist should promote desensitization (45). At least in the case of glycine receptors, the pore can also occupy wide-open conformations, inconsistent with open-state conductances let alone desensitization (46). To our knowledge, the possible contributions of sample preparation, classification methods, conformational heterogeneity, or other factors to this phenomenon remain to be elucidated; however, it seems likely to impact the determination and annotation of subtly distinct functional states in this receptor family for some time to come.

Indeed, multiple GLIC structures reported in this work were characterized by closed pores, including states consistent with either deprotonated or protonated conditions. It is theoretically possible that electrostatic conditions might be modified in cryo-EM by interaction with the glow-discharged grid or air–water interface, masking effects of protonation. However, we consistently noted subtle shifts in stability and conformation, indicating that local effects of protonation were reflected in the major resolved class. Several acidic residues could be confidently built only at lower pH; this enhancement could be a direct indicator of protonation, as anionic side chains have been reported to produce lower quality densities by cryo-EM (47, 48, 49). On the other hand, several

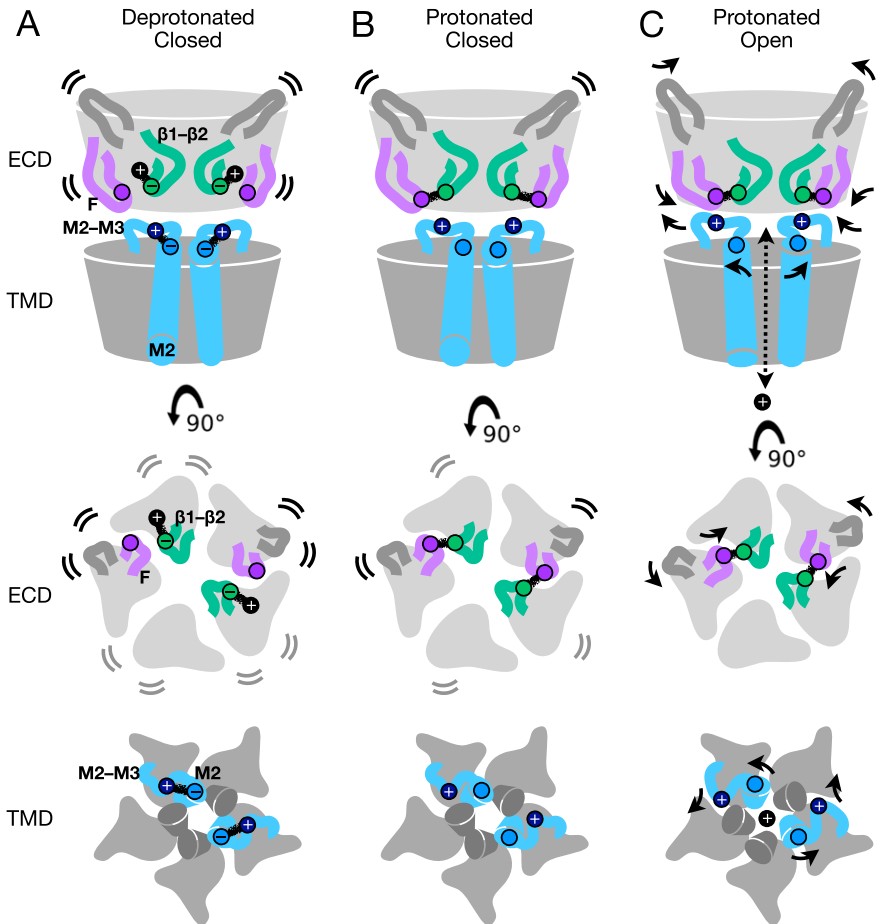

**Figure 5. Protonation and activation in GLIC pH gating.**
**(A)** Cartoon of the GLIC resting state, corresponding to a deprotonated closed conformation, as represented by the predominant cryo-EM structure at pH 7. Views are of the full protein (top) from the membrane plane, and of the ECD (middle) and TMD (bottom) from the extracellular side, showing key motifs at two opposing subunit interfaces including the principal β1–β2 (green) and M2–M3 loops (blue), complementary F (purple) and β5–β6 (dark gray) loops, and the remainder of the protein in light gray. By the model proposed here, under resting conditions the key acidic residue E35 (green circles) in the β1–β2 loop is deprotonated, and involved in transient interactions with environmental cations (e.g., Na⁺, black circles). Flexibility of the corresponding ECD is indicated by motion lines, associated with relatively low resolution by cryo-EM and high root mean-squared deviation ibn molecular dynamics simulations. In parallel, deprotonated E243 (light blue circles) in the M2 helix attracts K248 (dark blue circles) in the M2–M3 loop, maintaining a contracted upper pore. **(A, B)** Cartoon as in panel (A), showing a protonated but still closed conformation, as represented by the predominant cryo-EM structure at pH 3. In the ECD, protonation of E35 releases environmental cations and enables it instead to form a stabilizing contact with the complementary subunit via T158 (purple circles) in loop F, associated with partial rigidification of the ECD. In the TMD, protonation of E243 releases K248, allowing it to orient outward/upward towards the subunit/domain interface. **(A, C)** Cartoon as in panel (A), showing the putative protonated open state, as represented by previous open X-ray structures. Key side chains (E35, T158, E243, and K248) are arranged similar to the protonated closed state, accompanied by general contraction of the ECD including loop F, expansion of the upper TMD including the M2–M3 loop, and opening of the ion conduction pathway.

aspartate and glutamate residues were clearly defined in all our cryo-EM models, whereas others were poorly resolved in all conditions, indicating that factors such as local dynamics can play an even greater role; we therefore relied on side chain resolution along with other indicators to identify regions of focus in molecular simulations. Notably, the protonated closed state proposed here (Fig 5B) differs from previously reported locally closed and lipid-modulated forms, which have been captured for multiple GLIC variants at low pH (26, 28, 29, 30); the ECD in these structures is generally indistinguishable from that of the open state, suggesting the corresponding variations or modulators decouple extracellular transitions from pore opening (6, 36). In contrast, the minority class at pH 3 (state 2) approached open-state properties in both domains, including a contracted and untwisted ECD and a partly expanded pore (Figs 4 and S9).

The low-pH cryo-EM (state 1) structure features local rearrangements and dynamics consistent with protonation, for example, of residues E243/K248 and the M2–M3 loop, but with most of the protein backbone comparable to the resting state. Accordingly, this structure may correspond to a pre-open state on the opening pathway (36, 50, 51). The predominance of this state implies an open probability well below 100%, even under conditions shown to elicit maximal whole-cell currents (25). Due in part to low conductance, and technical challenges to measuring robust single-channel

currents under highly acidic conditions, efforts by multiple groups have yet to establish the maximal open probability for GLIC by electrophysiology (8, 52). However, we recently calculated an open probability of only 17% for GLIC at low pH, based on Markov state modeling of enhanced-sampling MD simulations (53 Preprint). Interestingly, low-pH crystallization of GLIC in lipidic mesophases also produced a closed structure, and a C-terminally tagged variant exhibited partial occupancy of both open and locally closed conformations in the same crystal form (32, 54): thus, even X-ray studies indicate a propensity for partial closure at low pH, and a role for crystal packing in influencing conformational selection. Most recently, we found small-angle neutron scattering of GLIC consistent with a low population of open channels at pH 3: although best-fit structures varied with pH, linear-combination fits to pH-3 scattering curves tolerated <20% contribution of open structures, indicating a heterogeneous mixture in solution-phase as well as cryo-EM conditions (55 Preprint).

An intriguing alternative is that this structure corresponds to a desensitized state, which would be expected to predominate at pH 3 subsequent to channel opening (34). However, previous spin-labeling studies found substantial differences between GLIC resting and desensitized states in the ECD and central pore (39, 52); in contrast, our pH-7 and pH-3 state-1 cryo-EM models were largely comparable in the ECD and pore backbones, with differences

primarily in interfacial side chain interactions and the M2–M3 loop. Indeed, desensitized states in this family are generally thought to resemble open states in the ECD and upper pore, but block conduction at a secondary intracellular-facing gate (6); in GLIC, such a mechanism is supported by the effect of divalent cations, which bind in the lower pore (9) and inhibit desensitization (34). Thus, although we cannot conclusively determine the physiological role of GLIC states resolved in this work, nor definitively quantify their representation in the mixture of states at a given pH, we propose that a pre-open conformation predominates in our low-pH samples.

The proton activation pathway described here may be a particular adaptation in GLIC; for most pentameric channels, agonist interactions are expected to be more localized to the orthosteric ligand-binding site. Still, increasing evidence indicates that a handful of titratable residues predominantly influence activation even in GLIC (11, 25). Furthermore, remodeling at the GLIC subunit/domain interface mirrors putative gating mechanisms in several other family members, including interdependent rearrangements in loop F and the β1–β2 and M2–M3 loops (2). Whereas loose packing of the ECD core was previously proposed as a gating strategy specific to eukaryotic members of this channel family (56), the parallels in GLIC are apparent: indeed, a heterogeneous mixture of closed states including an expanded ECD is consistent with previous atomic-force microscopy and X-ray crystallography studies (32, 35). Interestingly, recent cryo-EM structures of the zebrafish glycine receptor with partial agonists included minority classes corresponding to a so-called pre-flipped state, incorporating partial rearrangements of the ECD with a still-closed pore; thus, such metastable intermediates may contribute to gating even in eukaryotic family members (46). Furthermore, pH sensitivity has been reported for the bacterial channel sTeLIC as well as eukaryotic nicotinic, glycine, and GABA(A) receptors (57, 58, 59, 60), suggesting that proton sensing may be a conserved or convergent property of several family members.

The subtle dynamics of allosteric signal transduction in pentameric ligand-gated ion channels and their sensitivity to drug modulation have driven substantial interest in characterizing end point and intermediate structures along the gating pathway. Our data substantiate a protonated closed state, accompanied by a minority population with an expanded pore, and spotlight intrinsic challenges in capturing flexible conformations. We further offer a rationale for proton-stimulated side chain remodeling of multiple residues at key interfaces, with apparent parallels in other family members. Dissection of the gating landscape of a ligand-gated ion channel thus illuminates both insights and limitations of GLIC as a model system in this family, and support a mechanistic model in which entropy favors a flexible, expanded ECD, with agonists stabilizing rearrangements at the subunit/domain interface.

# Materials and Methods

### GLIC expression and purification

Expression and purification of GLIC-MBP was adapted from protocols published by Nury and colleagues (13). Briefly, C43(DE3)

*Escherichia coli* transformed with GLIC-MBP in pET-20b were cultured overnight at 37°C. Cells were inoculated 1:100 into 2xYT media with 100 µg/ml ampicillin, grown at 37°C to OD600 = 0.7, induced with 100 µM isopropyl-β-D-1-thiogalactopyranoside, and shaken overnight at 20°C. Membranes were harvested from cell pellets by sonication and ultracentrifugation in buffer A (300 mM NaCl and 20 mM Tris–HCl, pH 7.4) supplemented with 1 mg/ml lysozyme, 20 µg/ml DNase I, 5 mM $MgCl_2$, and protease inhibitors, then frozen or immediately solubilized in 2% n-dodecyl-β-D-maltoside (DDM). Fusion proteins were purified in batch by amylose affinity (NEB), eluting in buffer B (buffer A with 0.02% DDM) with 2–20 mM maltose, then further purified by size exclusion chromatography in buffer B. After overnight thrombin digestion, GLIC was isolated from its fusion partner by size exclusion, and concentrated to 3–5 mg/ml by centrifugation.

### Cryo-EM sample preparation and data acquisition

For freezing, Quantifoil 1.2/1.3 Cu 300 mesh grids (Quantifoil Micro Tools) were glow-discharged in methanol vapor before sample application. 3 µl sample was applied to each grid, which was then blotted for 1.5 s and plunge-frozen into liquid ethane using a FEI Vitrobot Mark IV. Micrographs were collected on an FEI Titan Krios 300 kV microscope with a post energy filter Gatan K2-Summit direct detector camera. Movies were collected at nominal 165,000× magnification, equivalent to a pixel spacing of 0.82 Å. A total dose of 40.8 $e^-/Å^2$ was used to collect 40 frames over 6 s, using a nominal defocus range covering −2.0 to −3.8 µm. This defocus range was selected to optimize signal-noise ratio/contrast, which was relatively low because of preferential distribution of particles in regions of very thick ice.

### Image processing

Motion correction was carried out with MotionCor2 (61). All subsequent processings were performed through the RELION 3.1 pipeline (62). Defocus was estimated from the motion corrected micrographs using CtfFind4 (63). After manual picking, initial 2D classification was performed to generate references for autopicking. Particles were extracted after autopicking, binned, and aligned to a 15-Å density generated from the GLIC crystal structure (PDB ID: 4HFI [16]) by 3D auto-refinement. The acquired alignment parameters were used to identify and remove aberrant particles and noise through multiple rounds of pre-aligned 2D- and 3D classification, which is particularly useful when dealing with small particles that might be hard to align or trying to focus on a particular region of interest (64, 65). In some cases, classes associated with fewer particles nonetheless resolved to higher resolution, and were chosen for subsequent refinement (Figs S1 and S10). The pruned set of particles was then refined, using the initially obtained reconstruction as reference. Per-particle CTF parameters were estimated from the resulting reconstruction using RELION 3.1. Global beam-tilt was estimated from the micrographs and correction applied. Micelle density was eventually subtracted and the final 3D auto-refinement was performed using a soft mask covering the protein, followed by post-processing, utilizing the same mask. Local resolution was estimated using the RELION implementation.

Post-processed densities were improved using ResolveCryoEM, a part of the PHENIX package (release 1.18 and later) (66) based on maximum-likelihood density modification, previously used to improve maps in X-ray crystallography (67). Densities from both RELION post-processing and ResolveCryoEM were used for building; figures show output from ResolveCryoEM (Figs 2 and S11A and B).

Densities for minority classes were obtained by systematic and extensive 3D classification in RELION 3.1, with iterative modifications including angular search, T parameter, and class number (Fig S10A–C). Our reported state-2 classes were at least partly dependent on classification parameters, emerging in the context of a large number of classes (50) and high-T parameter to amplify signal (100), extensive local searches applied during 3D classification and CTFs ignored until the first peak.

### Model building

Models were built starting from a template using an X-ray structure determined at pH 7 (PDB ID: 4NPQ (32), chain A), fitted to each reconstructed density. PHENIX 1.18.2–3874 (66) real-space refinement was used to refine this model, imposing fivefold symmetry through non-crystallographic symmetry restraints detected from the reconstructed cryo-EM map. The model was incrementally adjusted in COOT 0.8.9.1 EL (68) and re-refined until conventional quality metrics were optimized in agreement with the reconstruction. Model statistics are summarized in Table 1. Model alignments were performed using the match function in UCSF Chimera (69) on Cα atoms, excluding extracellular loops, for residues 17–192 (ECD) or 196–314 (TMD).

### MD simulations

Manually built cryo-EM structures, as well as previously published X-ray structures (resting, PDB ID: 4NPQ (32); open, PDB ID: 4HFI [16]), were used as starting models for MD simulations. Missing side chains were modeled with the WHAT IF server (70) without reference to previous structures. The Amber99sb-ILDN force field (71) was used to describe protein interactions. Each protein was embedded in a bilayer of 520 Berger (72) 1-palmitoyl-2-oleoyl-*sn*-glycero-3-phosphocholine lipids. Each system was solvated in a 14 × 14 × 15 nm$^3$ box using the TIP3P water model (73), and NaCl was added to bring the system to neutral charge and an ionic strength of 150 mM.

All simulations were performed with GROMACS 2019.3 (74). Systems were energy-minimized using the steepest descent algorithm, then relaxed for 100 ps in the NVT ensemble at 300 K using the velocity rescaling thermostat (75). Bond lengths were constrained (76), particle mesh Ewald long-range electrostatics used (77), and virtual sites for hydrogen atoms implemented, enabling a time step of 5 fs. Heavy atoms of the protein were restrained during relaxation, followed by another 45 ns of substance, pressure, temperature (NPT) relaxation at one bar using Parrinello-Rahman pressure coupling (78) and gradually releasing the restraints. Finally, to ensure conformational sampling particularly for poorly resolved residues, the system was relaxed with restraints only on Cα atoms of fully built residues for an additional 150 ns. For each relaxed system, four replicates of 1 µs unrestrained simulations were generated.

Analyses were performed using visual molecular dynamics (79), CHAP (38), and MDTraj (80), using values from the last 300 ns of each simulation. Time-dependent RMSDs were calculated for Cα atoms in generally resolved regions of the ECD (residues 15–48, 66–192) or TMD (residues 197–313). The number of sodium ions around E35 was quantified within a distance of 5 Å, using simulation frames sampled every 10 ns (400 total frames from four simulations in each condition), as described in Fig 3. PC analysis of the M2–M3 loop was performed on Cα atoms of residues E243–P250 of five superposed static models (three cryo-EM structures, resting and open X-ray structures), treating each subunit separately. The simulations were then projected onto PC1 (36% of the variance) versus PC2 (26% of the variance) and were plotted using kernel density estimation. Representative motions for PC1 and PC2 were visualized as sequences of snapshots from blue (negative values) to purple (positive values). As in previous work (37), ECD spread was determined by the radius of gyration around the channel axis of residues 20–190; domain twist was determined by the average dihedral angle defined by COM coordinates of (1) a single subunit-ECD, (2) the full ECD, (3) the full TMD, and (4) the same single-subunit TMD.

## Data Availability

Three-dimensional cryo-EM density maps of the pentameric ligand-gated ion channel GLIC in detergent micelles have been deposited in the Electron Microscopy Data Bank under accession numbers EMD-11202 (pH 7, state 1), EMD-12678 (pH 7, state 2), EMD-11208 (pH 5, state 1), EMD-12677 (pH 5, state 2), EMD-11209 (pH 3, state 1), and EMD-12675 (pH 3, state 2), respectively. Each deposition includes the cryo-EM sharpened and unsharpened maps, both half-maps and the mask used for final FSC calculation. Coordinates of state 1 models have been deposited in the Protein Data Bank. The accession numbers for the three GLIC structures are 6ZGD (pH 7), 6ZGJ (pH 5), and 6ZGK (pH 3). Full input data, parameters, settings, commands, and trajectory subsets from MD simulations are archived at Zenodo.org under DOI: 10.5281/zenodo.4320552.

## Supplementary Information

## Acknowledgements

The authors would like to thank the Swedish Cryo-EM National Facility staff, in particular Julian Conrad, José Miguel de la Rosa Trevin and Stefan Fleischmann from Stockholm and Michael Hall from Umeå, for kind assistance with data collection, modeling and supervision. This work was supported by grants from the Knut and Alice Wallenberg Foundation, the Swedish Research Council (2017-04641, 2018-06479, 2019-02433), the Swedish e-Science Research Centre, and the BioExcel Center of Excellence (EU 823830). U Rovšnik was supported by a scholarship from the Sven and Lilly Lawski Foundation. The cryo-EM data were collected at the Swedish national cryo-EM facility funded by the Knut and Alice Wallenberg Foundation, Erling

Persson and Kempe Foundations. Computational resources were provided by the Swedish National Infrastructure for Computing.

## Author Contributions

U Rovšnik: data curation, formal analysis, validation, investigation, visualization, and writing—original draft, review, and editing.

Y Zhuang: data curation, formal analysis, investigation, visualization, and writing—review and editing.

BO Forsberg: software, formal analysis, investigation, and writing—review and editing.

M Carroni: formal analysis, investigation, methodology, and writing—review and editing.

L Yvonnesdotter: formal analysis, investigation, methodology, and writing—review and editing.

RJ Howard: conceptualization, data curation, formal analysis, investigation, methodology, and writing—original draft, review, and editing.

E Lindahl: conceptualization, resources, software, funding acquisition, investigation, project administration, and writing—review and editing.

## Conflict of Interest Statement

The authors declare that they have no conflict of interest.

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
