## [Reviewer comments · Life Science Alliance]

Life Science Alliance

Dynamic closed states of a ligand-gated ion channel captured by cryo-EM and simulations

Urška Rovšnik, Yuxuan Zhuang, Björn Forsberg, Marta Carroni, Linnea Yvonnedotter, Rebecca Howard, and Erik Lindahl

DOI: <https://doi.org/10.26508/lsa.202101011>

Corresponding author(s): Erik Lindahl, Stockholm University and Rebecca Howard, Stockholm University

Review Timeline:

Submission Date:	2021-01-05
Editorial Decision:	2021-03-12
Revision Received:	2021-04-30
Editorial Decision:	2021-05-28
Revision Received:	2021-06-10
Accepted:	2021-06-11

Transaction Report:

March 12, 2021

Re: Life Science Alliance manuscript #LSA-2021-01011-T

Erik Lindahl
Stockholm University
Biochemistry and Biophysics
Science for Life Laboratory
Stockholm 17165
Sweden

Dear Dr. Lindahl,

Thank you for submitting your manuscript entitled "Dynamic closed states of a ligand-gated ion channel captured by cryo-EM and simulations" to Life Science Alliance. We apologize for this extended delay in getting back to you. The manuscript was assessed by expert reviewers, whose comments are appended to this letter.

As you will note from the reviewers' comments, while all 3 reviewers are excited about these findings, they have also raised some crucial points that must be addressed prior to further consideration of this manuscript at LSA. We, thus, encourage you to submit a revised version of the manuscript that addresses all of the reviewers' points.

Thank you for this interesting contribution to Life Science Alliance. We are looking forward to receiving your revised manuscript.

Sincerely,

Shachi Bhatt, Ph.D.

Executive Editor

Life Science Alliance

<https://www.lsjournal.org/>

Interested in an editorial career? EMBO Solutions is hiring a Scientific Editor to join the international Life Science Alliance team. Find out more here -

https://www.embo.org/documents/jobs/Vacancy_Notice_Scientific_editor_LSA.pdf

B. MANUSCRIPT ORGANIZATION AND FORMATTING:

Reviewer #1 (Comments to the Authors (Required)):

The manuscript by Rovsnik et al is a thought-provocative cryo-EM/MD-simulation study of the gating mechanism in pentameric ligand-gated ion channels (pLGIC). The object of the analysis is the prokaryotic channel GLIC, which is proton activated and whose structure at high resolution was

examined by cryo-EM at different pH conditions, i.e. at various concentrations of agonist. Based on both structural and simulation data, the most important conclusion is that protonation (i.e. agonist binding) and channel activation (i.e. opening of the channel) are distinct molecular events, such that a gating intermediate with high-affinity for the agonist and a closed ion pore would be functionally required. If so, this study would provide structural information at high resolution for this functional intermediate state.

Shedding light onto the gating mechanism in pLGICs is fundamentally important and pharmacologically relevant. In this context, studies like that of Rovsnik et al are timely and warmly welcome. In addition, this study provides unprecedented structural information and original conclusions that support a shift from the current mechanistic view. The manuscript is clearly written, the rationale is logical and easy to follow. However, I have some reservations concerning the main conclusion.

1. How physiologically relevant is the picture emerging from the cryo-EM analysis particularly at pH 3? From a functional point of view, electrophysiology has shown that GLIC is ion conducting at low pH. The present cryo-EM analysis shows that the vast majority of channels at these conditions are resting-like (state 1) and only a small fraction (state 2) resembles the active state. But, how small is the active-like fraction or in other words what is the probability of state 2 in these experiments? A careful analysis of the cryo-EM results similar to what was done in Yu, Gouaux et al bioRxiv 2019 may be used to address this question. Once done, would this fraction be consistent with the open probability of GLIC in single-channel recordings at physiological conditions and low pH? Although the authors state that the open probability of GLIC is not well established (page 10), this comparison is critical to reassure the reader about the physiological relevance of this study and its conclusions.

2. What is the evidence that the minority class at pH 3 actually corresponds to an active, ion-conducting state? As far as I can see, the only evidence is the quaternary change relative to state 1, i.e. some un-twisting and contraction of the ion channel (fig EV5), and the HOLE profile around 13' (fig 4D), which are both significantly different from those in the X-ray structure of GLIC at pH 4. Why these new cryo-EM structures should be physiologically more relevant than that captured by X-ray at low pH? Perhaps computational electrophysiology demonstrating cation permeability of state 2 may provide stronger support to the main conclusion. In the external electric-field setup, these calculations are straightforward and have been already used to explore ion conductance in pLGICs; see Cerdan et al, Structure 2018 or Damgen & Biggin, Structure 2019 as recent examples.

3. The generalization of the proposed mechanism to other pLGICs seems to me a bit far fetched. In fact, although one could understand the existence of a high-affinity, closed-channel state in the absence of a global conformational change in GLIC as a consequence of proton binding to multiple residues and subsequent side-chain rearrangements, it is harder to conceive the equivalent in eukaryotic homologues where agonist binding would be localized to the orthosteric site. Perhaps the Discussion on this point could be toned down.

4. Last, if the authors are correct in their reasoning, the reconstruction of the more twisted and contracted GLIC channel (state 2) would be critically important to explore gating, i.e. the conformational transition from closed to open, and should be provided as an output of this work, e.g. deposited in the PDB or made available.

Minor points

- In the Discussion (page 11), the authors state: "desensitized states in this family are generally thought to transition through an open state upon ligand dissociation, before returning to rest". In the absence of one or more references supporting this conclusion, this sentence cannot be used as an argument.

- I find that providing supporting material both in the Expanded View and the Appendix is very inconvenient. Since the distinction between the two is unclear, their co-existence is confusing and makes the supporting material less accessible to the reader.

Reviewer #2 (Comments to the Authors (Required)):

The manuscript reports on cryoEM data of GLIC obtained at different pHs, and on MD simulations performed on the corresponding structures (and also reference structures). It nicely dissects the role of charged residues involved in pH sensing, with complementary information gained from the experiments and from the simulations, thus providing a structural framework for previous functional studies.

An unexpected finding is that conformations at pH 3, 5 and 7 all seem compatible with closed states, while crystal structures of GLIC at low pHs feature wider pores compatible with an open state. A secondary state is seen at pH 3, which apparently has the hallmarks of an open-state.

The authors thus enrich our knowledge of the operation mechanism of GLIC with new structures (finally with cryoEM!), which are both informative and intriguing. The data is clearly presented and beautifully illustrated. I recommend this manuscript to be published. I also suggest altering the discussion on "which state does this closed structure correspond to?"

Those alteration may go along different lines:

While the authors are straightforward about what they observe, they miss the opportunity to relate this discrepancy between the observed structures and the expected states to the many other studies of pGLIC where it is also present. A non exhaustive list includes super open states of the GlyR, deemed desensitized state of the nAChRs that are actually wetted during simulations, non-desensitized states of the 5-HT₃R in the presence of agonists, etc...

In my humble opinion, discussing some of those cases in relation to the observed data would strengthen the paper by reminding the reader of other cases where the structure-to-state assignment was delicate.

The two-fold argument on why the observed pH3 state is not desensitized looked a bit meager and might be expanded. Is there any mutagenesis data on GLIC that indicates that the D gate is intracellular? Is the EPR data on GLIC (ref 27, 35, Chakrapani lab) rather in agreement with the observed structure not being desensitized?

I may have missed it, but the % of particles being in the minority conformation at pH 3 should be stated. It's not clear if it corresponds to the 1'648 particle class depicted in EV1, and the methods are not clear either (and should be modified, see minor comments).

Many questions came to my mind while reading about this minority conformation, and the text could address some of them:

Could it be obtained with better res if collecting more data? (I'm NOT asking for it)? With authors in a lab next to a Krios, I guess they would have done it if it were possible

Are methods looking at variability, such as 3DVA analysis in Cryosparc, able to pick a closed-to-open mode? Is that also present at pH 5?

How are its M2s exactly? we do not see it in Figures

Why not deposit the map, instead of saying it's available on request

This would be my major point: while the authors may rightfully be reluctant to put too much emphasis on a low resolution map, they should provide more information about the second state. I'm not sure I understand the argument about how the present structures explain the predominance of open-pore crystal structures at low pH. If crystal contact interfaces are particularly important, maybe the authors could strengthen it by showing they can't pack their closed structure using the crystal packing (=there are clashes)

Minor comments

Abstract

"These results support a dissection of protonation and activation steps..."

I'm not a native speaker, so it might just be my limited English skills, but I am still not sure what to get from this sentence. That the protonation and activation steps can be uncoupled or sequential? or that the results permit an in-depth analysis of those protonation and activation steps?

Main text

Page 4:

"indicating that differential resolution could not be trivially attributed to data quantity"

I wonder whether this test with equivalent numbers of randomly chosen particles tells anything about data quality. But the wording is very prudent, and that may be the best that one can do to compare data sets, so very ok.

After reading this sentence I went to the table and saw that the attrition between picked and kept particles was different for the pH 7 dataset compared to the two others (~700k picked, 86k kept). The authors may comment on that in the text, about possible reasons for so many particles being excluded, about what's wrong with the 97'151 class shown in Fig. EV1....

In other words: isn't this attrition another hint of the higher flexibility of the structure at high pH? (and finally, similar to my comment on the minority state, it would be informative to see a 3DVA movie of a consensus refinement of particles at pH7)

"Whereas loose packing of the ECD core has been proposed as a gating strategy specific to eukaryotic members of this channel family [43]; our data indicate an expanded, flexible ECD may also be important to earlier evolutionary branches"

Maybe I'm mistaken, but wasn't loose packing of ECD in the closed state(s) one of the main findings of GLIC high pH crystal structure? The motion referred as blooming/unblooming in ref 33

If yes, this manuscript's data confirms/expands previous findings, with the advantage of having particles not constrained by a crystal packing.

It might also be discussed whether this is eukaryotic vs prokaryotic, or pH versus binding of a molecule. pH will act on many residues, it's a signal distributed onto the whole structure while neurotransmitter binding may require an orthosteric site not too deformed by ECD flexibility.

Methods

Not sure I understand "multiple rounds of pre-aligned 2D- and 3D-classification". Are those

classifications without realignments? If yes, I ignored it was common to do that in 2D and thus a sentence explaining why or a ref might be useful

"Densities for minority classes were obtained by systematic and extensive 3D-classification rounds in RELION 3.1, with iterative modifications to parameters including angular search, T parameter, and class number"

Ok, it's normal to do lots of classification trials, and the reader do not need all the details, but what the text does not tell is whether the secondary state was always appearing in those classifications, or whether it was a rare event that required a unique combination of parameters.

Simple curiosity, why is the defocus range starting at -2 microns and not a more classical value, such as -0.7/-1 ?

Figures

EV5

The 'spread' and the 'twist' graphed in figure EV5 are those of the experimental structures. It would be a plus to see the values for the corresponding simulations.

Figure 4

I find it hard to picture the opening of the M2s in pH3 state 2 versus state 1. There should be an additional panel that show densities, e.g. in EV5, top superimposed view, showing TMDs. Figure 4 is not sufficient here, especially as the views is 4b are from 'the outside' and one cannot see M2s well.

Reviewer #3 (Comments to the Authors (Required)):

This paper provides novel cryo-EM structures of a much-studied prokaryotic. In doing so, it offers previously unseen info on conformational changes during channel gating and offers several hypotheses for future experimental testing.

Two major points

I'm not a cryo-EM expert, but I would be interested to see the maps before making final comments on the submission. Unless I've missed something in the files, we currently get only a glimpse of the density for selected segments in two EV/supporting figures. (Hence the "Maybe" selection under "Experimental Evidence".)

The crux of the paper is the rearrangement of carboxylate side chains during conformational changes of GLIC in protonation. Unfortunately, the side chains of carboxylates are not perfectly to study via cryo-EM (detailed below). This doesn't negate the results/discussion, but how this was dealt with in modeling and simulations should be clearer.

Details/minor comments

page 5 - "[D31's] enhanced definition further supported stabilization of the beta1-beta2 loop". I don't think it's fair to say that, if a side chain is unresolved in one model and resolved in another, that the its appearance in the latter stabilizes the latter structure. Perhaps it's doing the same

thing in the other structure too, but it's simply unresolved.

page 6 text - It's suggested that side chain flexibility/Na⁺ binding makes E35 flexible and therefore unresolved. Isn't it possible that the charged carboxylate side chain is not resolved (and low pH, protonated carboxylates are resolved) due to inherent issues of cryo-EM and such side chains? e.g. Vonck & Mills, 2017, Curr Op Struc Biol; Wang, 2017, Prot Sci.

page 6 text - How was the side chain of E35 modeled for the starting position in the resting state simulation? (Simply based on 4NPQ and then relaxed?) Would this affect the likelihood of Na⁺ binding, as opposed to e.g. interactions with vicinal protein atoms?

pages 6-7 - The text here and the PC analysis suggest "stretching of the loop across" an interface. Does this mean that the simulations suggested that the M2-M3 loop moved more noticeably than as implicated in a comparison of static models at pH 7/5/3, where rearrangements were minimal (Fig 2)? Or does this refer to side chains only? Perhaps a description of overall rearrangements (main chain as well as side chain), not just RMSD, for the simulations would help here, before getting into the interpretations/side chain discussions.

Page 7 - We read about additionally structure classes. I can't see how these derive from the workflow in Fig EV1. Could this be clarified?

The rearrangement of Loop F in a putative pre-active state is interesting. Did this not appear dynamic in the simulations (page 6)?

Page 9 middle paragraph - E243 or E343?

Erik Lindahl
Professor of Biophysics

Dear Dr. Bhatt,

Thank you for the positive comments about our work; we appreciate the reviewers' thoughtful and thought-provoking remarks. We believe we have responded to all concerns, with comments and responses itemized below for ease of reference in the comparison document. We have also deposited three additional reconstructions in the EMDB, added two supporting figures (Figs S6, S10) and edited two others (Figs S7, S9), removed one supporting image (former Fig S5), and generated three videos for reviewer transparency, linked in Responses 2.4 and 2.M3.

Below, all reviewer comments are **gray**, with our text (including *Comment* and *Response* itemization) in **black**, and quotes from the manuscript in **purple**. Changes to the manuscript are marked in a separate document using the track-changes function in MS Word, including references to relevant reviewer comments; page numbers in this letter correspond to the tracked-changes version.

Reviewer #1 (Comments to the Authors (Required)):

The manuscript by Rovsniik et al is a thought-provocative cryo-EM/MD-simulation study of the gating mechanism in pentameric ligand-gated ion channels (pLGIC). The object of the analysis is the prokaryotic channel GLIC, which is proton activated and whose structure at high resolution was examined by cryo-EM at different pH conditions, i.e. at various concentrations of agonist. Based on both structural and simulation data, the most important conclusion is that protonation (i.e. agonist binding) and channel activation (i.e. opening of the channel) are distinct molecular events, such that a gating intermediate with high-affinity for the agonist and a closed ion pore would be functionally required. If so, this study would provide structural information at high resolution for this functional intermediate state.

Shedding light onto the gating mechanism in pLGICs is fundamentally important and pharmacologically relevant. In this context, studies like that of Rovsniik et al are timely and warmly welcome. In addition, this study provides unprecedented structural information and original conclusions that support a shift from the current mechanistic view. The manuscript is clearly written, the rationale is logical and easy to follow. However, I have some reservations concerning the main conclusion.

Comment 1.1: How physiologically relevant is the picture emerging from the cryo-EM analysis particularly at pH 3? From a functional point of view, electrophysiology has shown that GLIC is ion conducting at low pH. The present cryo-EM analysis shows that the vast majority of channels at these conditions are resting-like (state 1) and only a small fraction (state 2) resembles the active state. But, how small is the active-like fraction or in other words what is the probability of state 2 in these experiments? A careful analysis of the cryo-EM results similar to what was done in Yu, Gouaux et al bioRxiv 2019 may be used to address this question. Once done, would this fraction be consistent with the open probability of GLIC in single-channel recordings at physiological conditions and low pH? Although the authors state that the open probability of GLIC is not well established (page 10), this comparison is critical to reassure the reader about the physiological relevance of this study and its conclusions.

Department of Biochemistry and Biophysics

Stockholm University
Science for Life Laboratory
Box 1031
SE-171 21 Solna, Sweden

Visiting address:
Science for Life Laboratory
Tomtebodavägen 23A, Solna

Phone: +46-(0)73-4618050
Cell: +46734618050
Mail: erik.lindahl@dbb.su.se

Response 1.1: The reviewer raises an important point about the distribution of conducting and nonconducting states; we wrestled with the utility of classification proportions, as employed by Yu et al. and others, in accurately representing state distributions in our samples. Given the evident interest in this question, we have now **expanded our results** (page 7) to specify that state 2 at pH 3 represents ~23,000 particles, 3.3% of the initially extracted particles from the full dataset; in contrast, state 1 represents 46%. Particle counts are now also available for all state-2 models in the newly **added Fig S10**, in addition to counts already available for state-1 models in Fig S1. To explain our interpretation of these findings, we have also **expanded our discussion** (page 10) to review some caveats in extrapolating from particle numbers to state distributions:

“This state was associated with 3.3% of particles used for 3D classification at pH 3, with state 1 accounting for 46% (23,000 vs 214,000 out of 690,000, Figs S10 and S1); however, we refrained from attributing particle counts directly to state distributions. To our knowledge, all cryo-EM classification methods are stochastic optimization tools subject to local minima, and not well suited to inferring quantitative ensemble statistics, particularly when the number and functional annotation of total states remains unclear.”

The reviewer’s point about comparing open probabilities is also well taken; we have accordingly **added the following text, and several new references** with alternative quantifications, to the discussion of apparent Popen (page 12):

“The predominance of this state implies an open probability well below 100%, even under conditions shown to elicit maximal whole-cell currents [25]. Due in part to low conductance, and technical challenges to measuring robust single-channel currents under highly acidic conditions, efforts by multiple groups have yet to establish the maximal open probability for GLIC by electrophysiology [8], [52]. However, we recently calculated an open probability of only 17% for GLIC at low pH, based on Markov state modeling of enhanced-sampling MD simulations [53]. Interestingly, low-pH crystallization of GLIC in lipidic mesophases also produced a closed structure, and a C-terminally tagged variant exhibited partial occupancy of both open and locally closed conformations in the same crystal form [32], [54]: thus, even X-ray studies indicate a propensity for partial closure at low pH, and a role for crystal packing in influencing conformational selection. Most recently, we found small-angle neutron scattering of GLIC consistent with a low population of open channels at pH 3: although best-fit structures varied with pH, linear-combination fits to pH-3 scattering curves tolerated less than 20% contribution of open structures, indicating a heterogeneous mixture in solution-phase as well as cryo-EM conditions [55].”

Comment 1.2: What is the evidence that the minority class at pH 3 actually corresponds to an active, ion-conducting state? As far as I can see, the only evidence is the quaternary change relative to state 1, i.e. some un-twisting and contraction of the ion channel (fig EV5), and the HOLE profile around 13' (fig 4D), which are both significantly different from those in the X-ray structure of GLIC at pH 4. Why these new cryo-EM structures should be physiologically more relevant than that captured by X-ray at low pH? Perhaps computational electrophysiology demonstrating cation permeability of state 2 may provide stronger support to the main conclusion. In the external electric-field setup, these calculations are straightforward and have been already used to explore ion conductance in pLGICs; see Cerdan et al, Structure 2018 or Damgen & Biggin, Structure 2019 as recent examples.

Response 1.2: These comments helpfully highlight a possible miscommunication in our original work, where we sought to define pH-3 state 2 as a potential intermediate on the pathway towards opening. This state is clearly distinct from state 1, but in the absence of a higher-resolution reconstruction we cannot be confident of its atomistic structure, dynamics, hydration, or conductive behavior, and therefore intended to refrain from interpreting it as an open state. To clarify this point, **we have rephrased several sentences** in the final section of the results (pages 7–8), particularly modifying the final clause from “...activating transitions at low pH” to “...transition towards an open state at low pH.” Moreover, we have **added the following sentence to the discussion** (page 10):

“Thus, although the low resolution of our minority-class reconstructions precluded atomistic

modeling or quantitative simulations of permeability, our pH-3 minority class could represent an intermediate or alternative along the GLIC gating pathway.”

Comment 1.3: The generalization of the proposed mechanism to other pLGICs seems to me a bit far fetched. In fact, although one could understand the existence of a high-affinity, closed-channel state in the absence of a global conformational change in GLIC as a consequence of proton binding to multiple residues and subsequent side-chain rearrangements, it is harder to conceive the equivalent in eukaryotic homologues where agonist binding would be localized to the orthosteric site. Perhaps the Discussion on this point could be toned down.

Response 1.3: This point is well taken, also reflected by *Comment 2.M4*. To limit potentially spurious extrapolations, **we have removed Fig S5 and some related discussion** of interfacial loop remodeling (~13 lines of text). Instead, **we have revised the penultimate paragraph** of the discussion (page 13) to focus on specific evidence of comparable gating mechanisms in e.g. GABA(A) and glycine receptors:

“The proton activation pathway described here may be a particular adaptation in GLIC; for most pentameric channels, agonist interactions are expected to be more localized to the orthosteric ligand-binding site. Still, increasing evidence indicates that a handful of titratable residues predominantly influence activation even in GLIC [11] [25]. Furthermore, remodeling at the GLIC subunit/domain interface mirrors putative gating mechanisms in several other family members, including interdependent rearrangements in loop F and the $\beta 1$ – $\beta 2$ and M2–M3 loops [2]. Whereas loose packing of the ECD core was previously proposed as a gating strategy specific to eukaryotic members of this channel family [56], the parallels in GLIC are apparent: indeed, a heterogeneous mixture of closed states including an expanded ECD is consistent with previous atomic-force microscopy [35] and X-ray crystallography studies [32]. Interestingly, recent cryo-EM structures of the zebrafish glycine receptor with partial agonists included minority classes corresponding to a so-called pre-flipped state, incorporating partial rearrangements of the ECD with a still-closed pore; thus, such metastable intermediates may contribute to gating even in eukaryotic family members [53]. Furthermore, pH sensitivity has been reported for the bacterial channel sTeLIC as well as eukaryotic nicotinic, glycine, and GABA(A) receptors [57]–[60], suggesting that proton sensing may be a conserved or convergent property of several family members.”

Comment 1.4: Last, if the authors are correct in their reasoning, the reconstruction of the more twisted and contracted GLIC channel (state 2) would be critically important to explore gating, i.e. the conformational transition from closed to open, and should be provided as an output of this work, e.g. deposited in the PDB or made available.

Response 1.4: With thanks for this important point (also mentioned by reviewers 2 (*Comment 2.4*) and 3 (*Comment 3.1*)), **state-2 reconstructions at pH 7, pH 5, and pH 3 have now been deposited in the EMDB**, as indicated in Data Availability (accession numbers EMD-12678, EMD-12677, and EMD-12675 respectively).

Minor points

Comment 1.M1: In the Discussion (page 11), the authors state: "desensitized states in this family are generally thought to transition through an open state upon ligand dissociation, before returning to rest". In the absence of one or more references supporting this conclusion, this sentence cannot be used as an argument.

Response 1.M1: We have now **revised and more thoroughly referenced our discussion of desensitization** (pages 12–13); the full text of the new paragraph is quoted in response to reviewer 2 (*Comment 2.2*), who raised a similar concern.

Comment 1.M2: I find that providing supporting material both in the Expanded View and the Appendix is very inconvenient. Since the distinction between the two is unclear, their coexistence is confusing and makes the supporting material less accessible to the reader.

Response 1.M2: Point taken. **We have now classified all supporting figures as Supplementary Material**, and renumbered them according to their appearance in the text (Figs S1–S11).

Reviewer #2 (Comments to the Authors (Required)):

The manuscript reports on cryoEM data of GLIC obtained at different pHs, and on MD simulations performed on the corresponding structures (and also reference structures). It nicely dissects the role of charged residues involved in pH sensing, with complementary information gained from the experiments and from the simulations, thus providing a structural framework for previous functional studies.

An unexpected finding is that conformations at pH 3, 5 and 7 all seem compatible with closed states, while crystal structures of GLIC at low pHs feature wider pores compatible with an open state. A secondary state is seen at pH 3, which apparently has the hallmarks of an open-state.

The authors thus enrich our knowledge of the operation mechanism of GLIC with new structures (finally with cryoEM!), which are both informative and intriguing. The data is clearly presented and beautifully illustrated. I recommend this manuscript to be published. I also suggest altering the discussion on "which state does this closed structure correspond to?"

Those alteration may go along different lines:

Comment 2.1: While the authors are straightforward about what they observe, they miss the opportunity to relate this discrepancy between the observed structures and the expected states to the many other studies of pGLIC where it is also present. A non exhaustive list includes super open states of the GlyR, deemed desensitized state of the nAChRs that are actually wetted during simulations, non-desensitized states of the 5-HT3R in the presence of agonists, etc...

In my humble opinion, discussing some of those cases in relation to the observed data would strengthen the paper by reminding the reader of other cases where the structure-to-state assignment was delicate.

Response 2.1: We appreciate this stimulating suggestion, and have **added a new paragraph to the discussion** (pages 10–11) as follows:

“Interestingly, the growing catalog of pentameric ligand-gated ion-channel structures now includes several examples in which experimental data—largely from X-ray crystallography or cryo-EM—correspond to unanticipated functional states. In the case of ELIC, all structures to date have been incompatible with conduction even in the presence of saturating agonist, possibly due to a lack of critical lipid interactions [42]-[44]. Some eukaryotic channels exhibit an opposite tendency, adopting hydrated or open states even where prolonged exposure to agonist should promote desensitization [45]. At least in the case of glycine receptors, the pore can also occupy wide-open conformations, inconsistent with open-state conductances let alone desensitization [46]. To our knowledge, the possible contributions of sample preparation, classification methods, conformational heterogeneity, or other factors to this phenomenon remain to be elucidated; however, it seems likely to impact the determination and annotation of subtly distinct functional states in this receptor family for some time to come.”

Comment 2.2: The two-fold argument on why the observed pH3 state is not desensitized looked a bit meager and might be expanded. Is there any mutagenesis data on GLIC that indicates that the D gate is

intracellular? Is the EPR data on GLIC (ref 27, 35, Chakrapani lab) rather in agreement with the observed structure not being desensitized?

Response 2.2: As noted also in *Response 1.M1*, we have now **modified our discussion of desensitization** (pages 10, 12–13) to better characterize our rationale, particularly with reference to EPR data and other evidence for an intracellular desensitization gate:

“A proposed desensitized-GLIC structure determined by co-crystallization with the inhibitor docosahexaenoic acid also contained a partly-contracted upper pore [26], supporting the presence of alternative conformations in the conduction pathway...

“An intriguing alternative is that this structure corresponds to a desensitized state, which would be expected to predominate at pH 3 subsequent to channel opening [34]. However, previous spin-labeling studies found substantial differences between GLIC resting and desensitized states in the ECD and central pore [39], [52]; in contrast, our pH-7 and pH-3 state-1 cryo-EM models were largely comparable in the ECD and pore backbones, with differences primarily in interfacial sidechain interactions and the M2–M3 loop. Indeed, desensitized states in this family are generally thought to resemble open states in the ECD and upper pore, but block conduction at a secondary intracellular-facing gate [6]; in GLIC, such a mechanism is supported by the effect of divalent cations, which bind in the lower pore [9] and inhibit desensitization [34]. Thus, we propose that a pre-open conformation predominates in our low-pH samples, although other states cannot be excluded.”

Comment 2.3: I may have missed it, but the % of particles being in the minority conformation at pH 3 should be stated. It's not clear if it corresponds to the 1'648 particle class depicted in EV1, and the methods are not clear either (and should be modified, see minor comments).

Response: We regret the confusion our processing figures may have caused, also raised by reviewer 1 (*Comment 1.1*); we have now **added Fig S10** with pipelines for all three alternative states, and **edited the methods** (page 16) as described in *Comment 2.M6*.

Comment 2.4: Many questions came to my mind while reading about this minority conformation, and the text could address some of them:

Could it be obtained with better res if collecting more data? (I'm NOT asking for it)? With authors in a lab next to a Krios, I guess they would have done it if it were possible

Are methods looking at variability, such as 3DVA analysis in Cryosparc, able to pick a closed-to-open mode? Is that also present at pH 5?

How are its M2s exactly? we do not see it in Figures

Why not deposit the map, instead of saying it's available on request

This would be my major point: while the authors may rightfully be reluctant to put too much emphasis on a low resolution map, they should provide more information about the second state.

Response 2.4: As the reviewer has correctly assumed, we are eager to improve the resolution particularly of the minority conformation at pH 3. However, we have observed through processing of this and other systems, as well as internal development work in particle classification, that reconstruction quality does not necessarily scale with input particles. Based on the extensive classification efforts (now documented in our **added Fig S10**) required to distinguish our reported minority states, we concluded that even a substantial investment in additional data collection would not assuredly improve the particular class identified here as pH-3 state 2, and was beyond the scope of the current work.

As the reviewer also helpfully suggests, we have also performed 3D-variability analysis using the tool in cryoSPARC on datasets in all three pH conditions. As expected, we observed some variability at pH 3 particularly related to expansion/contraction of the ECD; however, dominant modes did not clearly visualize opening transitions of the channel pore (**video 1**, <https://bit.ly/3aM1B7M>). At pH 5 (**video 2**, <https://bit.ly/2Sh1r25>) and pH 7 (**video 3**, <https://bit.ly/3aO8ICb>), dominant modes of variability were even less clearly related to anticipated transitions.

Although these results intriguingly support differential variability at low pH reminiscent of gating, due to the absence of a definitive pore mode we have focused our present edits on clarifying the

comparisons of state-1 and state-2 reconstructions, adding a subpanel to **Fig S9A** focused on M2 in the TMD.

We have also **deposited maps for all three** minority-class states, as also requested by reviewers 1 and 3 (*Comments 1.4, 3.1*).

Comment 2.5: I'm not sure I understand the argument about how the present structures explain the predominance of open-pore crystal structures at low pH. If crystal contact interfaces are particularly important, maybe the authors could strengthen it by showing they can't pack their closed structure using the crystal packing (=there are clashes)

Response 2.5: With thanks for this suggestion, we have added a figure (**Fig S6**) illustrating the clash of all three majority-class cryo-EM densities particularly in the upper ECD with symmetry partners that would exist in the open-state crystal (PDB ID 4HFI), and **modified the results** as follows (page 4):

“Interestingly, superposition of our cryo-EM reconstructions with equivalent molecules in an open-state GLIC crystal lattice revealed clashes particularly at the outer periphery of the ECD (Fig S6), indicating packing effects that could influence conformational selection.”

Minor comments

Comment 2.M1: Abstract - "These results support a dissection of protonation and activation steps..." I'm not a native speaker, so it might just be my limited English skills, but I am still not sure what to get from this sentence. That the protonation and activation steps can be uncoupled or sequential ? or that the results permit an in-depth analysis of those protonation and activation steps?

Response 2.M1: **This sentence has been revised** to read “These results allow us to define distinct protonation and activation steps...”

Comment 2.M2: Main text, page 4 - "indicating that differential resolution could not be trivially attributed to data quantity"

I wonder whether this test with equivalent numbers of randomly chosen particles tells anything about data quality. But the wording is very prudent, and that may be the best that one can do to compare data sets, so very ok.

Response 2.M2: **The statement has been expanded in results** (page 4) to more explicitly reflect uncertainties in this analysis: “This comparison indicates that differential resolution could not be trivially attributed to data quantity, although it does not identify the cause of differential data quality.”

Comment 2.M3: After reading this sentence I went to the table and saw that the attrition between picked and kept particles was different for the pH 7 dataset compared to the two others (~700k picked, 86k kept). The authors may comment on that in the text, about possible reasons for so many particles being excluded, about what's wrong with the 97'151 class shown in Fig. EV1...

In other words: isn't this attrition another hint of the higher flexibility of the structure at high pH? (and finally, similar to my comment on the minority state, it would be informative to see a 3DVA movie of a consensus refinement of particles at pH7)

Response 2.M3: As the reviewer rightly notices, in some cases sizable classes have been excluded from further analysis. For instance, the 97k-class only resolved to 6.4 Å resolution, indicating that associated particles were relatively heterogeneous or low-quality compared to the class chosen for subsequent processing. The reviewer also raises an interesting question as to whether apparent particle heterogeneity or quality can be used as evidence for higher flexibility, at least in the case of our pH-7 data; however, as also indicated in *Response 2.4*, our 3D-variability analysis of this dataset (**video 3**, <https://bit.ly/3aO8ICb>) did not demonstrate a mode we could confidently attribute to domain- or state-dependent flexibility. We have therefore limited ourselves to **clarifying our classification rationale** in methods (page 15): “In some cases, classes associated with fewer particles nonetheless resolved to higher resolution, and were chosen for subsequent refinement (Figs S1, S10).”

Comment 2.M4: "Whereas loose packing of the ECD core has been proposed as a gating strategy specific to eukaryotic members of this channel family [43]; our data indicate an expanded, flexible ECD may also be important to earlier evolutionary branches"

Maybe I'm mistaken, but wasn't loose packing of ECD in the closed state(s) one of the main findings of GLIC high pH crystal structure? The motion referred as blooming/unblooming in ref 33

If yes, this manuscript's data confirms/expands previous findings, with the advantage of having particles not constrained by a crystal packing.

It might also be discussed whether this is eukaryotic vs prokaryotic, or pH versus binding of a molecule. pH will act on many residues, it's a signal distributed onto the whole structure while neurotransmitter binding may require an orthosteric site not too deformed by ECD flexibility.

Response 2.M4: With thanks for these comments, we have **revised the corresponding section of discussion** (page 13); these changes were also noted in *Response 1.3*, but are partially restated here due to their relevance:

"The proton activation pathway described here may be a particular adaptation in GLIC; for most pentameric channels, agonist interactions are expected to be more localized to the orthosteric ligand-binding site. Still, increasing evidence indicates that a handful of titratable residues predominantly influences activation even in GLIC [11], [25]. Furthermore, remodeling at the GLIC subunit/domain interface mirrors putative gating mechanisms in several other family members, including interdependent rearrangements in loop F and the $\beta 1$ – $\beta 2$ and M2–M3 loops [2]. Whereas loose packing of the ECD core was previously proposed as a gating strategy specific to eukaryotic members of this channel family [56], the parallels in GLIC are apparent: indeed, a heterogeneous mixture of closed states including an expanded ECD is consistent with previous atomic-force microscopy and X-ray crystallography studies [32], [35]."

Comment 2.M5: Methods - Not sure I understand "multiple rounds of pre-aligned 2D- and 3D-classification". Are those classifications without realignments? If yes, I ignored it was common to do that in 2D and thus a sentence explaining why or a ref might be useful.

Response 2.M5: We have **expanded this section of methods** (page 15), including **two relevant references**: "The acquired alignment parameters were used to identify and remove aberrant particles and noise through multiple rounds of pre-aligned 2D- and 3D-classification, which is particularly useful when dealing with small particles that might be hard to align or trying to focus on a particular region of interest [64], [65]."

Comment 2.M6: "Densities for minority classes were obtained by systematic and extensive 3D-classification rounds in RELION 3.1, with iterative modifications to parameters including angular search, T parameter, and class number"

Ok, it's normal to do lots of classification trials, and the reader do not need all the details, but what the text does not tell is whether the secondary state was always appearing in those classifications, or whether it was a rare event that required a unique combination of parameters.

Response 2.M6: As the reviewer has correctly assumed, state-2 reconstructions emerged in the context of an unusually large number of classes, as well as an increased T-parameter to amplify signal. It also appeared critical to apply extensive local searches during 3D classification, and to ignore CTFs until the first peak. In addition to showing these processing pipelines in the new figure (**Fig S10**) mentioned in *Comments 1.1* and 2.3, we have now **extended this portion of the methods** (page 16):

"Densities for the minority classes were obtained by systematic and extensive 3D-classification in RELION 3.1, with iterative modifications including angular search, T- parameter, and class number (Fig S10). The emergence of reported state-2 classes was at least partly dependent on classification parameters, emerging in the context of a large number of classes (50) and high T-parameter to amplify signal (100), with extensive local searches applied during 3D classification and CTFs ignored until the first peak."

Comment 2.M7: Simple curiosity, why is the defocus range starting at -2 microns and not a more classical value, such as -0.7/-1 ?

Response 2.M7: This is an interesting point; we agree that -0.7 would be a more classical and preferable defocus. However, particularly our early grid preparations only contained appreciable particles in regions of thick ice, and accordingly low contrast, requiring the use of higher defocus values. We have now **extended the methods** (page 15):

“This defocus range was selected to optimize signal-noise ratio/contrast, which was relatively low due to preferential distribution of particles in regions of very thick ice.”

Comment 2.M8: Figures - EV5 - The 'spread' and the 'twist' graphed in figure EV5 are those of the experimental structures. It would be a plus to see the values for the corresponding simulations.

Response 2.M8: The median and 25% confidence intervals for **relevant MD simulations have now been added to Fig S9B** (former Fig EV5), and described in the corresponding legend. The **methods used to calculate these values have also been rephrased** (page 17):

“As in previous work [38], ECD spread was determined by the radius of gyration around the channel axis of residues 20 to 190; domain twist was determined by the average dihedral angle defined by COM coordinates of 1) a single subunit-ECD, 2) the full ECD, 3) the full TMD, and 4) the same single-subunit TMD.”

Comment 2.M9: Figure 4 - I find it hard to picture the opening of the M2s in pH3 state 2 versus state 1. There should be an additional panel that show densities, e.g. in EV5, top superimposed view, showing TMDs. Figure 4 is not sufficient here, especially as the views in 4b are from 'the outside' and one cannot see M2s well.

Response 2.M9: To better address this, we have added the requested subpanel to **Figure S9A** (formerly EV5), also described in *Response 2.4*.

Reviewer #3 (Comments to the Authors (Required)):

This paper provides novel cryo-EM structures of a much-studied prokaryotic. In doing so, it offers previously unseen info on conformational changes during channel gating and offers several hypotheses for future experimental testing.

Two major points

Comment 3.1: I'm not a cryo-EM expert, but I would be interested to see the maps before making final comments on the submission. Unless I've missed something in the files, we currently get only a glimpse of the density for selected segments in two EV/supporting figures. (Hence the "Maybe" selection under "Experimental Evidence".)

Response 3.1: As described in *Responses 1.4* and *2.4*, **we have now deposited the requested maps**, and cited them under Data Availability.

Comment 3.2: The crux of the paper is the rearrangement of carboxylate side chains during conformational changes of GLIC in protonation. Unfortunately, the side chains of carboxylates are not perfectly to study via cryo-EM (detailed below). This doesn't negate the results/discussion, but how this was dealt with in modeling and simulations should be clearer.

Response 3.2: The reviewer rightly emphasizes the challenge of interpreting differential sidechain resolution, particularly for acidic groups. Both older (Holger et al., 1996) and more recent (Hattne et al., 2018) electron-diffraction studies have investigated the effects of electron damage on different types of bonds, and definitively, acidic chains are susceptible to decarboxylation. On the other hand, as

reported by Vonck and Mills (2017, also mentioned by the reviewer) D and E residues are also subject to radiation damage in X-ray crystallography, but no one would doubt the information given from crystallographic electron densities about these amino acids. Indeed, more and more structures in the cryo-EM literature include perfectly visible D and E residues in certain regions, and not in others; here for example, D185 and E272 have completely visible side-chains, while D88 is consistently poorly defined. Ultimately, we used sidechain resolution along with other indicators of local remodeling to form our mechanistic hypotheses, which we further examined using molecular dynamics. To clarify these points, **we have added these citations and extended our consideration of the issue in reference to Comments 3.M1–3.M3 below.**

Details/minor comments

Comment 3.M1: page 5 - "[D31's] enhanced definition further supported stabilization of the beta1-beta2 loop". I don't think it's fair to say that, if a side chain is unresolved in one model and resolved in another, that its appearance in the latter stabilizes the latter structure. Perhaps it's doing the same thing in the other structure too, but it's simply unresolved.

Response 3.M1: With thanks for the alternative interpretation, we have **edited this section of results** to emphasize that these sidechain differences served primarily as a basis for subsequent MD analysis (page 5):

“At the midpoint of the same $\beta 1$ – $\beta 2$ loop, density surrounding basic residue K33 was similarly absent at pH 7 and pH 5, but clearly defined a sidechain oriented down towards the TMD at pH 3 (Fig 2B). An additional $\beta 1$ – $\beta 2$ acidic residue, D31, could also be uniquely built at pH 3, oriented in towards the central vestibule. Enhanced definition of these sidechains could indicate a relatively uniform state of this loop at pH 3, improved cryo-EM signal for protonated versus anionic acidic groups, or other factors; dynamics in this region were accordingly explored in the next section.”

Comment 3.M2: page 6 text - It's suggested that side chain flexibility/Na⁺ binding makes E35 flexible and therefore unresolved. Isn't it possible that the charged carboxylate side chain is not resolved (and low pH, protonated carboxylates are resolved) due to inherent issues of cryo-EM and such side chains? e.g. Vonck & Mills, 2017, Curr Op Struc Biol; Wang, 2017, Prot Sci.

Response 3.M2: As indicated in *Response 3.2*, we cannot exclude effects of radiation damage on E35 at higher pH, although our subsequent simulation results persuaded us of an additional or alternative influence of interactions with environmental ions vs neighboring protein regions. As further described in *Response 3.M1*, **we have now rephrased the description of results** to make this possibility more explicit; we also **edited the discussion** (page 11) to include the two suggested references:

“Several acidic residues could be confidently built only at lower pH; this enhancement could be a direct indicator of protonation, as anionic sidechains have been reported to produce lower-quality densities by cryo-EM [47]-[49]. On the other hand, several D and E residues were clearly defined in all our cryo-EM models, while others were poorly resolved in all conditions, indicating that factors such as local dynamics can play an even greater role; we therefore relied on sidechain resolution along with other indicators to identify regions of focus in molecular simulations.”

Comment 3.M3: page 6 text - How was the side chain of E35 modeled for the starting position in the resting state simulation? (Simply based on 4NPQ and then relaxed?) Would this affect the likelihood of Na⁺ binding, as opposed to e.g. interactions with vicinal protein atoms?

Response 3.M3: We have now **expanded the methods section** (page 16) to clarify that unbuilt residues were modeled de novo: “Missing sidechains were modeled with the WHAT IF server [70] without reference to previous structures.” Relevant to Na⁺ versus protein binding, we further employed a lengthy equilibration specifically to the unbuilt sidechains; however, we recognize the implications of this step may not have been clear. We have now **emphasized later in the methods** (page 17) that “...to refine conformations particularly for poorly resolved sidechains, the system was relaxed with restraints only on C α atoms of fully built residues for an additional 150 ns. [...] Analyses

were performed using VMD [79], CHAP [38], and MDTraj [80], using data from the last 300 ns of each simulation.”

Comment 3.M4: pages 6-7 - The text here and the PC analysis suggest "stretching of the loop across" an interface. Does this mean that the simulations suggested that the M2-M3 loop moved more noticeably than as implicated in a comparison of static models at pH 7/5/3, where rearrangements were minimal (Fig 2)? Or does this refer to side chains only? Perhaps a description of overall rearrangements (main chain as well as side chain), not just RMSD, for the simulations would help here, before getting into the interpretations/side chain discussions.

Response 3.M4: We regret the lack of clarity in our initial loop analysis, and have now **emphasized that both PCs were based on comparisons of C α traces**, not sidechains; thus, the loop “stretching” indeed refers to backbone transitions evident in simulations, that may not be apparent in our static-model figures (page 7):

“To explore larger-scale rearrangements in M2–M3 that might accompany these sidechain transitions, we further plotted the two lowest principal components (PC) of C α atom cartesian coordinate covariance in the M2–M3 loop, revealing distinct conformation distributions in simulations of pH-7, pH-5, and pH-3 models (Fig 3F). The two dominant PCs for this motif were associated with flipping of the central loop (C α atoms near K248) from a downward to outward orientation (PC1), and stretching of the loop backbone across the subunit/domain interface (PC2).”

As suggested, we have also now **outlined overall rearrangements alongside RMSD in the first paragraph of MD results** (page 6): “Beyond these domain-level deviations, distinct transitions were observed with decreasing pH in a subset of sidechain and backbone atoms at the ECD subunit interface and in the M2–M3 loop, as described below (Figs 3, S7C).”

Comment 3.M5: Page 7 - We read about additionally structure classes. I can't see how these derive from the workflow in Fig EV1. Could this be clarified?

Response 3.M5: As also suggested in *Comments 1.1* and *2.M6*, **we have added Fig S10** to show the independent processing pipelines used to obtain minority states.

Comment 3.M6: The rearrangement of Loop F in a putative pre-active state is interesting. Did this not appear dynamic in the simulations (page 6)?

Response 3.M6: To elaborate on the flexibility of loop F, we have now **added a histogram** for loop-F C α RMSD (Fig S8C), and **the following sentence to results** (page 6):

“Interestingly, loop F—including T158—exhibited lower C α deviations in simulations of the pH-3 versus higher-pH models, a pattern also exhibited by low- versus high-pH X-ray structures (Fig S8C).”

Comment 3.M7: Page 9 middle paragraph - E243 or E343?

Response 3.M7: Thank you for pointing out this typing error; **the correct designation (E243) is now in place.**

Once again, thank you so much for the valuable comments and help in improving the work!

Sincerely,

/Erik Lindahl/

May 28, 2021

RE: Life Science Alliance Manuscript #LSA-2021-01011-TR

Prof. Erik Lindahl
Stockholm University
Biochemistry and Biophysics
Science for Life Laboratory
Stockholm 17165
Sweden

Dear Dr. Lindahl,

Thank you for submitting your revised manuscript entitled "Dynamic closed states of a ligand-gated ion channel captured by cryo-EM and simulations". We would be happy to publish your paper in Life Science Alliance pending final text revisions necessary to address Reviewer 1's remaining concern and to meet our formatting guidelines.

Please also attend to the following:

- please add Keywords for your manuscript in our system
- please add ORCID ID for the corresponding author-you should have received instructions on how to do so
- please add your table legend to the main manuscript text after the main and supplementary figure legends;
- please add callouts for Figures S1A-C; S2A-C; S3A-C; S5A-C; S7B; S10A-C to your main manuscript text
- please add scale bar for Figure 5B
- Scale bar for figure 5A is not visible enough - please improve readability

A. FINAL FILES:

B. MANUSCRIPT ORGANIZATION AND FORMATTING:

Sincerely,

Shachi Bhatt, Ph.D.
Executive Editor
Life Science Alliance
<http://www.lsajournal.org>
Tweet @SciBhatt @LSAJournal

Reviewer #1 (Comments to the Authors (Required)):

I would like to thank the authors for their careful consideration of my comments. Overall, they have done a good job and I am convinced the manuscript has improved during the review process. From my perspective, there is only one important point left, which remains to be tackled.

Response 1.2: Following the authors reasoning, it is concluded that state 2 at pH=3 is not open, rather it represents a potential closed-channel intermediate on the pathway towards opening. Since state 1 appears to be inconsistent with desensitization and is structurally consistent with the resting state at pH=7 (see Response 2.2), I wonder what is the effect of agonist binding in the samples collected at pH=3? In fact, if the open probability at saturating conditions is about 20% and there is no evidence of gating in these samples (no channel opening, nor desensitization) how can be claimed that the experiments at pH=3 are physiologically relevant? I believe, the authors should provide a more convincing argument.

Reviewer #2 (Comments to the Authors (Required)):

The revised manuscript by Rovsnik et al. -together with the extensive rebutall that accompanies the manuscript- fully adresses the issues and comments made during the first round of review.

The paper now has a well-balanced discussion of what the second state is and how it relates to physiological states. The added data (new deposited maps), technical details and illustrations make it look very good overall!

I don't have additional requests.

Reviewer #3 (Comments to the Authors (Required)):

The changes outlined by the authors satisfy my initial concerns.

I cannot seem to find the described CryoEM maps at EPDB, but the response letter seems to address my concerns.

Erik Lindahl
Professor of Biophysics

Dear Dr. Bhatt,

Thank you for the positive decision. We believe we have addressed the last remaining items:

Reviewer #1:

Following the authors reasoning, it is concluded that state 2 at pH=3 is not open, rather it represents a potential closed-channel intermediate on the pathway towards opening. Since state 1 appears to be inconsistent with desensitization and is structurally consistent with the resting state at pH=7 (see Response 2.2), I wonder what is the effect of agonist binding in the samples collected at pH=3? In fact, if the open probability at saturating conditions is about 20% and there is no evidence of gating in these samples (no channel opening, nor desensitization) how can be claimed that the experiments at pH=3 are physiologically relevant? I believe, the authors should provide a more convincing argument.

With regret for the confusion, we have further edited the discussion (p. 12 l. 16–21) to clarify that state 1 at pH 3 is not identical to the resting state:

'The low-pH cryo-EM (state 1) structure features local rearrangements and dynamics consistent with protonation, e.g. of residues E243/K248 and the M2–M3 loop, but with most of the protein backbone comparable to the resting state. Accordingly, this ~~With a resting-like backbone configuration, but sidechains consistent with proton activation, the low-pH cryo-EM (state 1) structure may correspond to a pre-open state on the opening pathway [36], [50], [51].'~~

We have also expanded discussion of the data processing (p. 11 l. 4–6), to clarify that the fact that our majority classes contain closed or intermediate pores should not be interpreted as an absence of open states: **'Indeed, additional conformations including an open state are likely present in our samples, but were not clearly distinguished among predominant classes of well ordered particles.'**

Finally, we have expanded the penultimate page of discussion (p. 13 l. 17–21) to acknowledge that we cannot definitively confirm the physiological role of our resolved pH-3 states, nor their extent of representation in the mixture of states at pH 3:

'Thus although we cannot conclusively determine the physiological role of GLIC states resolved in this work, nor definitively quantify their representation in the mixture of states at a given pH, we propose that a pre-open conformation predominates in our low-pH samples, ~~although other states cannot be excluded.'~~

Reviewer #3:

I cannot seem to find the described CryoEM maps at EPDB, but the response letter seems to address my concerns.

With apologies for the lack of clarity, the maps have been deposited on EMDB, but are currently held for publication. If necessary we are happy to provide the map files, though it appears the reviewer did not require this.

We have also added the requested keywords, ORCID, table legend, figure callouts and modified the scale bars in Figure S1.

Sincerely,

/Erik Lindahl/

Department of Biochemistry and Biophysics

Stockholm University
Science for Life Laboratory
Box 1031
SE-171 21 Solna, Sweden

Visiting address:
Science for Life Laboratory
Tomtebodavägen 23A, Solna

Phone: +46-(0)73-4618050
Cell: +46734618050
Mail: erik.lindahl@dbb.su.se

June 11, 2021

RE: Life Science Alliance Manuscript #LSA-2021-01011-TRR

Prof. Erik Lindahl
Stockholm University
Biochemistry and Biophysics
Science for Life Laboratory
Stockholm 17165
Sweden

Dear Dr. Lindahl,

Thank you for submitting your Research Article entitled "Dynamic closed states of a ligand-gated ion channel captured by cryo-EM and simulations". It is a pleasure to let you know that your manuscript is now accepted for publication in Life Science Alliance. Congratulations on this interesting work.

DISTRIBUTION OF MATERIALS:

Again, congratulations on a very nice paper. I hope you found the review process to be constructive and are pleased with how the manuscript was handled editorially. We look forward to future exciting submissions from your lab.

Sincerely,
